

# Distribution, trends and drivers of flash droughts in the United Kingdom.

**Ivan Noguera.[1], Jamie Hannaford.[1, 2], Maliko Tanguy [1,3]**

[1]UK Centre for Ecology & Hydrology (UKCEH), Wallingford, United Kingdom

[2]Irish Climate Analysis and Research UnitS (ICARUS), Maynooth University, Maynooth, Ireland

[3]European Centre for Medium-Range Weather Forecasts

Correspondence: Ivan Noguera (ivanog@ceh.ac.uk). UK Centre for Ecology & Hydrology, Maclean Building, Benson Lane, Crowmarsh Gifford, Wallingford, Oxon, OX10 8BB, UK.

## Abstract

Flash droughts have been the subject of a great deal of scientific attention in the last decade, but the greatest emphasis has been on relatively dry climates. Here, we characterised the occurrence of this type of rapid-onset drought events in a more humid setting, the United Kingdom (UK), for the period 1969-2021. Our results show that flash droughts affected both the wetter regions of north-west and the drier regions of south-east in every season over the last five decades. However, the spatiotemporal distribution of flash droughts is highly variable in UK, with important regional and seasonal contrasts. Central and northern regions were generally the most frequently affected by flash droughts in comparison to southeastern region. Overall, there are non-significant trends in flash drought frequencies in winter, summer, and autumn. Nevertheless, we found a significant and notable increase in the number of flash droughts recorded in spring months. In the UK, flash drought occurrence responds primarily to precipitation variability in all seasons, and particularly in winter and autumn. In spring and summer, the atmospheric evaporative demand (AED) is important as a secondary driver for triggering flash droughts, especially in the drier regions of the southeastern UK. Moreover, our findings evidenced that this relevance is rising significantly in spring and summer in the southeast, over the study period. The atmospheric and oceanic conditions controlling these anomalies in precipitation and AED that drive flash droughts were also



analyzed. Remarkable anomalies in sea level pressure and 500 hPa geopotential height
associated with the presence of high-pressure systems were noted over UK during the
development of the most severe flash droughts in all seasons. Likewise, flash drought
development typically occurred under negative phase of North Atlantic Oscillation phase
in winter and autumn, while in summer and spring positive phase is dominant. We also
found positive anomalies in sea surface temperature during the development of flash
droughts in spring and summer, while mixed anomalies were reported in winter and
autumn. This study presents a detailed characterisation of flash drought phenomenon in
UK, providing useful information for drought assessment and management, and a
climatology of flash droughts that can be used as a baseline against which future changes
in flash drought occurrence can be assessed.
**Keywords:** flash drought, precipitation deficit, atmospheric evaporative demand (AED),
Standardized Precipitation Evapotranspiration Index (SPEI), ocean-atmospheric
conditions, North Atlantic Oscillation (NAO), United Kingdom.

## 48   1. Introduction

Drought is one of the most damaging natural hazards worldwide, with major
impacts on natural and socioeconomic systems (Mishra and Singh, 2010; Wilhite, 2000;
Wilhite and Glantz, 1985). It is also widely regarded as a very complex phenomenon –
its development is usually slow, cascading through the diverse sectors affected in periods
that range from months to years (Wilhite and Pulwarty, 2017). However, recent studies
have demonstrated that some droughts events, commonly termed as "flash droughts", can
develop at much shorter timescales (Otkin et al., 2018). Flash droughts are distinguished
by an unusually rapid development associated with severe precipitation deficits that are
often accompanied by increases in atmospheric evaporative demand (AED) associated,
for example, with wave episodes (Pendergrass et al., 2020). Such rapid-onset drought
events affects both humid and dry regions, causing important agriculture and environment
impacts, particularly alongside elevated temperatures – including rapid decreases in soil
moisture that result in agricultural stress and increase the risk of wildfires, and rapid
declines in river flow that trigger impacts on aquatic wildlife (e.g. fish kills) and water
quality problems like algal blooms, as well as localized challenges in meeting public
water supply. In addition, flash droughts pose particular challenge for decision-making
and drought management and communication, given their rapid onset (Otkin et al., 2022).





Nowadays, the study of flash droughts has become a topic of great interest to the

scientific community and water managers. Many efforts have been made in recent years
to analyse flash drought phenomena using different approaches based on very diverse
metrics (e.g. soil moisture, AED, precipitation, climatic water balance, etc.) (Lisonbee et
al., 2021). Thus, several studies have assessed this phenomenon in various regions of the
world over the last two decades from different perspectives (Walker et al., 2023). Despite
progress in the understanding of this phenomenon, there are still many issues that are
poorly understood, particularly those related to the drivers and mechanisms involved in
triggering flash droughts. This is further complicated by the large seasonal and spatial
variations in the characteristics of flash droughts, particularly marked between water-
limited (i.e. dry areas characterised by a low water availability) and energy-limited (i.e.
humid areas characterised by a high water availability) regions (Mukherjee and Mishra,
2022; Noguera et al., 2021). Furthermore, most of the literature on flash droughts focuses
exclusively on a few regions (i.e. primarily United States and China) (Christian et al.,
2024). As a result, there are still important gaps in the regional knowledge of flash drought
characteristics in many regions of the world.

This study focuses on the United Kingdom (UK), a temperate oceanic, mild and

mostly humid region characterised by a predominance of energy-limited conditions
(Hulme and Barrow, 1997; Mayes and Wheeler, 1997), but with significant variations
including some more water-limited areas in the south-east (Kay et al., 2013) – an area
with a particularly fine balance between water supply and demand that already
experiences significant water stress (Folland et al., 2015). Hence, while the UK is
generally regarded as a wet country, it is regularly affected by severe droughts with major
agricultural, hydrological, and environmental impacts (Barker et al., 2019; Pribyl, 2020;
Spraggs et al., 2015).

Many studies have analysed drought phenomena in the UK, including; spatial

and temporal characterisation (Burke and Brown, 2010; Rahiz and New, 2012; Tanguy et
al., 2021), propagation through the hydrological cycle (Barker et al., 2016; Folland et al.,
2015) or drought impact assessment on different environmental and socioeconomic
systems (Byers et al., 2020; Dobson et al., 2020; Parsons et al., 2019), among others.
However, most of drought studies in UK are focused on long times scales (e.g 12-
months), while droughts developing at short-term have had comparatively little attention.



In this way, no studies previously analysed specifically the occurrence of flash droughts
in UK.
Most severe droughts are commonly related to long-term precipitation deficits
(Marsh et al., 2007; Todd et al., 2013; Barker et al. 2019), but notable increases in AED
at short-term can be essential in explaining the rapid development and aggravation of
some extreme droughts. In recent decades, several drought events strongly driven by rises
in AED (e.g. associated with heat waves episodes) were reported (Wreford and Neil
Adger, 2010). Some studies broadly distinguish between 'multiannual' droughts that
primarily affect southeast England (e.g. 2004 – 2006; 2010 – 2012), and within-year
'summer' droughts that can affect all areas (e.g. 1995, 2003) (Barker et al., 2019; Marsh
et al., 2007). Many droughts are in fact a combination of these 'types'. It is certainly the
case that some of the most testing historical droughts, including the 'benchmark' 1976
drought, have involved heatwave conditions associated with very high AED. Recent
examples include the 2018 and 2022 summer drought (Barker et al., 2024; Turner et al.,
2021), which caused severe impacts on fluvial and terrestrial ecosystems, water supply
or crop yields as a result of a lack in precipitation that was exacerbated by rapid increases
in AED.
Under climate change, numerous studies suggest a general increase in drought
severity (Dai, 2011; Vicente-Serrano et al., 2022) associated with the rise of AED at
global scale (Vicente-Serrano et al., 2020; Wang et al., 2012). In this direction, some
studies focused on flash drought showed an increase of this kind of events in different
regions around the world related to global warming (Mishra et al., 2021; Noguera et al.,
2022; Wang and Yuan, 2021; Yuan et al., 2018, 2019). In UK, various studies suggest an
increase in drought frequency and severity (Rahiz and New, 2013; Reyniers et al., 2023),
as well as the impacts associated with these hydroclimatic events (Gosling, 2014; Richter
and Semenov, 2005) as a consequence of climate change. While there is significant
uncertainty in future projections of how multiannual droughts will evolve in future (Lane
and Kay, 2021), future projections of hotter, drier summers suggest a high likelihood in
the increase in more widespread (Tanguy et al., 2023) within-year summer droughts
(Parry et al., 2024), and with this, likely increases flash droughts. Before such future
changes can be quantified, there is a need to understand an observational baseline of flash
drought occurrence and identify any emerging trends.





The greatest attention on flash droughts has been in dry (i.e. water-limited)
regions as flash droughts are, intuitively, expected to have less impact in humid regions
such as UK due to perceived high water availability– noting, as discussed, that in reality
parts of the south-east are relatively dry and subject to tangible water stresses. Moreover,
while they may be intuitively less prevalent the occurrence of flash droughts can also
have very severe implications and their frequency and severity may also increase under
global warming. Therefore, it is needed to understand the characteristics of flash drought
in these regions, as well as unravel the process and mechanisms controlling its occurrence.
The UK climate is complex, with different synoptic mechanisms operating at different
spatial scales, but also by the strong ocean-atmosphere interactions and the orographic
configuration in the region (Mayes and Wheeler, 2013). Among others, the strong
influence of large-scale drivers such as North Atlantic Oscillation (NAO) is well-known
for controlling climate variability over the UK, especially in northern and western regions
and during winter months (Fowler and Kilsby, 2002; Lavers et al., 2010; Murphy and
Washington, 2001; West et al., 2019, 2021b). Some studies have also shown that other
large-scale circulation patterns such as the East Atlantic Pattern, Scandinavian pattern
play a secondary role in modulating precipitation in UK (Bueh and Nakamura, 2007;
Hannaford et al., 2011; Ummenhofer et al., 2017; West et al., 2021a), while there is also
an underlying role for slowly-varying modes of ocean-atmosphere variability such as the
Atlantic Multidecadal Oscillation and ENSO (Folland et al., 2015; Svensson and
Hannaford, 2019). While there is a good general understanding of these mechanisms in
driving rainfall variability, their role in droughts is complex, and hence there is a gap in
understanding of the drivers of both multi-annual and short-term flash droughts.
In this study, we present a detailed characterisation of the flash drought
phenomenon in the UK, making the first (to the authors' knowledge) comprehensive,
national-scale analysis of flash droughts in this region- and one which can serve as a
testbed for other relatively wet locations which may expect to see increases in flash
drought severity in future. To achieve this purpose, we address several objectives: i) to
characterise the spatial and temporal occurrence of flash droughts over the UK; ii) to
analyse the observed trends in their frequency over the last five decades; iii) to assess the
role of the different meteorological factors involved in this type of drought events; and
iv) to identify the atmospheric and oceanic conditions under which flash droughts
develop.





## 2. Data and methods

### 2.1 Meteorological data

We employed gridded precipitation and potential evaporative (PET) data with high spatial and temporal resolution for the UK in the period 1969-2021. On the one hand, precipitation daily data at 1km$^2$ was obtained from Met Office Hadley Centre for Climate Science and Services (Met Office, 2018). All details on the creation and validation of the gridded precipitation data are provided by Hollis et al. (2019). On the other hand, PET daily data at 1km$^2$ was obtained from Environmental Information Data Centre (EIDC) (Brown et al., 2023). PET data was obtained from maximum and minimum air temperature, relative humidity, sunshine duration, and wind speed by means of Penman-Monteith equation, providing a robust metric of atmospheric evaporative demand (AED). Additional details about the creation, validation, and computation of gridded dataset in (Robinson et al., 2023). Daily information of precipitation and AED was aggregated weekly to calculate the climatic water balance (i.e. difference between precipitation and AED), which was employed to obtain the Standardized Precipitation Evapotranspiration Index (SPEI) (Vicente-Serrano et al., 2010).

### 2.2 Flash drought identification

We used the SPEI to identify flash droughts as it is sensitive to the variability of precipitation and AED (Tomas-Burguera et al., 2020), thus considering the main meteorological drivers of flash droughts triggering. SPEI is based on the standardisation of the difference between precipitation and AED (i.e. climatic water balance), providing comparable values in time and space (Beguería et al., 2014). In addition, SPEI is a multiscalar index that allows to fit computation time scale to the temporal response of the diverse natural and human systems affected by drought. Thus, many studies have used SPEI to analyse the response of hydrological (Lorenzo-Lacruz et al., 2010; Peña-Gallardo et al., 2019a; Vicente-Serrano and López-Moreno, 2005), agricultural (Peña-Gallardo et al., 2018b, 2019b; Potop et al., 2012) and environmental (Peña-Gallardo et al., 2018a; Vicente-Serrano et al., 2013, 2014; Zhang et al., 2017) systems to drought. Moreover, several studies have also demonstrated the good performance of SPEI for flash drought assessment (Hunt et al., 2014; Noguera et al., 2020, 2021).

As suggested by Noguera et al. (2020), we employed the SPEI at a short time scale (1-month) and high frequency (weekly) to identify rapid and anomalous changes in





humidity conditions associated with flash drought onset (Otkin et al., 2018; Svoboda et
al., 2002). Thus, a flash drought is defined as a decline in SPEI values equal to or less
than -2 z-units over a four-week period (i.e. development phase) that ends in a SPEI value
equal to or less than -1.28 z-units (corresponding to a return period of 10 years). The four-
week period established for the development phase allows the metric to capture rapid
variations in humidity conditions, but which persist long enough to expect some impact
(Noguera et al., 2020), which is consistent with the most widely used definitions for the
assessment of flash droughts (Anderson et al., 2013; Chen et al., 2019; Christian et al.,
2019; Osman et al., 2020; Mukherjee and Mishra, 2022). Applying this definition, we
identified all flash drought events that occurred in UK over the period 1969-2021 at
seasonal scale (winter: DJF, spring: MAM, summer: JJA, autumn: SON), as well as for
the growing-season (MAMJJAS). Further details of the method employed to identify
flash drought events can be found in Noguera et al. (2020).
Given the large climatic differences across the UK, we carry out flash drought
analysis at regional scale. There is a strong southeast-northwest gradient in precipitation
across the UK, with values ranging from >3000mm to <600mm annually (Mayes and
Wheeler, 2013). This strong gradient results in important differences between the drought
patterns observed in the wetter northwestern and the drier southeastern regions. In order
to assess the possible regional differences in flash drought characteristics, we considered
three regions: North-West, Transition and South-East (Figure A1). The regional division
used here is derived from Tanguy et al. (2021), who used a k-mean clustering technique
to divide the UK into three regions based on long-term (1862-2015) precipitation patterns.
This delineates a wetter (i.e. North-West) and a drier region (i.e. South-East), as well as
a transitional region (Transition) between both. Since flash droughts are primarily driven
by precipitation variability (Hoffmann et al., 2021; Koster et al., 2019), it is expected to
be the most important factor controlling their characteristics and spatiotemporal
behaviour in the UK.
**2.3 Assessment of the AED contribution**
To unravel the contribution of AED to SPEI we calculated the index allowing
precipitation to vary according to the observed climate evolution, while the AED
remained at its mean value, which was set at the average AED for each week of the year
over the period 1969–2021. This version of the index (hereafter referred to as SPEI_PRE)



that only responds to precipitation variations was compared with the original SPEI series.
In order to determine the relative contribution of AED to the development of flash
droughts, we considered that the difference between zero and SPEI_PRE was due to
precipitation variability, while the difference between SPEI_PRE and SPEI was due to
the contribution of AED. The differences were expressed as percentages, and for those
weekly data in which SPEI_PRE was equal to or less than SPEI, the AED contribution
was considered 0%. This type of approach has been used in numerous studies to calculate
the relative contribution of different variables in triggering drought conditions (Cook et
al., 2014; Noguera et al., 2022; Scheff and Frierson, 2014; Williams et al., 2015; Zhao
and Dai, 2015).
Given that our objective is to analyse the role of the AED as a driver of flash
drought development, we examined the contribution of the AED in the weekly data
corresponding to the onset of each of the flash drought events identified, as it captures the
cumulative anomaly in the climatic balance over the four-week period of the development
phase. Furthermore, we specifically analysed the spatial and temporal patterns of the
AED contribution to the development of flash droughts for the three regions considered
and on a seasonal scale over the period 1969–2021.
**2.4 Atmospheric and oceanic data**
To analyse the atmospheric mechanism underlying flash drought ocurrence in
UK, we focused on atmospheric conditions recorded during the development phase (i.e.
the four-week prior to flash drought onset). In order to show a set of events representative
of the atmospheric conditions typically associated with the triggering of flash droughts,
we focus on the events with the largest area affected. For this purpose, we selected the
top-10 flash droughts identified in each season (winter: DJF, spring: MAM, summer: JJA
and autumn: SON) for the period 1969-2021 according to the percentage of the UK area
affected in a given week.
We employed daily sea level pressure (SLP) and 500 hPa geopotential height
(Z500) data obtained from the National Centers for Environmental Prediction (NCEP)–
National Center for Atmospheric Research (NCAR) reanalysis (Kalnay et al., 1996) for
the domain study (25ºN-70ºN, 45ºW-45ºE) over the period 1969-2021 at 5º spatial
resolution. To illustrate the synoptic situations associated with flash drought, we
calculated SLP and Z500 anomalies during the development of the top-10 flash droughts





identified in each. The anomalies are relative to the average SLP and Z500 over the period
1969-2021. We also evaluated the possible seasonal relationship between flash drought
ocurrence the most important large-scale circulation patterns affecting UK: North
Atlantic Oscillation (NAO). For this purpose, we calculated NAO index (NAOi)
following the approach proposed by Jones et al. (1997), which is based on the differences
between normalised SLP at the points 36°N, 5°W (Gibraltar, United Kingdom) and 65°N,
20°W (Reykjavik, Iceland). Then, we computed the average anomalies recorded in NAOi
during the development of the top-10 flash droughts identified in each season over the
period 1969-2021.
To examine the possible conection between the development of flash droughts
and oceanic conditions, we analysed sea surface temperature (SST) anomalies during the
development phase of the top-10 flash droughts identified in each season (winter: DJF,
spring: MAM, summer: JJA and autumn: SON) for the period 1982-2021 according to
the percentage of the UK area affected in a given week. We employed daily SST
anomalies data obtained from the National Centers for Environmental Prediction
(NCEP)–National Center for Atmospheric Research (NCAR) reanalysis for the domain
study (25ºN-70ºN, 45ºW-45ºE) over the period 1982-2021 at 0.25º spatial resolution. In
this case, we focus on the period 1982-2021, instead of the period 1969-2021, given the
temporal availability of the data.
**2.5 Trends calculation**
We examined magnitude of change in flash drought frequencies using a linear
regression analysis between the time series (independent variable) and the time series of
flash droughts (dependent variable). We also employed this approach to calculate the
seasonal magnitude of change in Precipitation, AED, and AED contribution to flash
drought development. Then, to assess the significance of the trends over the period 1969-
2021, we employed the nonparametric Mann-Kendall statistic. Autocorrelation was
included in the trend analysis using the modified Mann-Kendall trend test, which returned
corrected p-values after accounting for temporal pseudoreplication (Hamed and
Ramachandra Rao, 1998; Yue and Wang, 2004).
**3. Results**
**3.1 Spatial distribution and trends**



The spatial distribution of flash droughts in the UK shows a large seasonal
variability in the UK, as well as important regional differences (Figure 1). In winter, the
highest number of flash droughts was recorded in Northern Ireland and central UK, while
the south coast and northeastern region reported the lowest number of events. Large areas
along north-south of UK and Northern Ireland were highly affected by flash droughts in
spring, with more than 15 events reported over the study period. By contrast, southeastern
and northwestern regions are generally least affected by flash droughts during the spring.
A clear gradient in the number of flash droughts was noted in summer, with important
variations from the southeast, where a low number of flash droughts are found (5-10
events), to the northwest of UK, which recorded the highest number of events. In autumn,
Northern Ireland and southwestern region were more frequently affected by flash
droughts, whereas southeastern and northeastern regions reported the lower occurrence
of events.

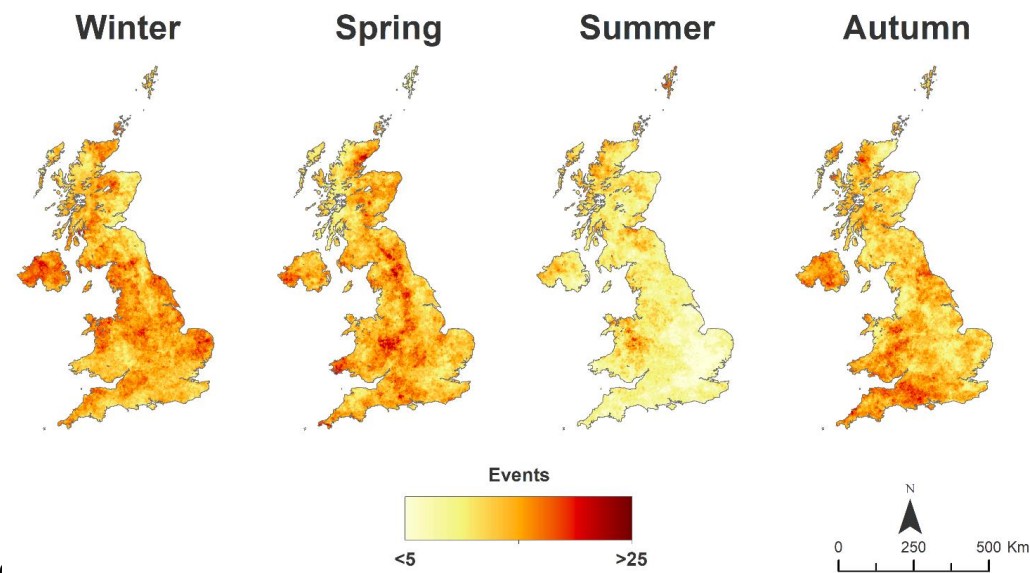

**Figure 1.** Seasonal spatial distribution of the total number of flash droughts in United
Kingdom for the period 1969-2021.

Focusing on growing-season, when the impacts associated to flash drought are
expected to be greater, it is possible to recognized large areas affected by flash droughts
along north-south of UK (Figure 2). Among others, the west of UK and Northern Ireland
were the most affected areas, with more than 35 events recorded. Whereas southeastern
UK were the least frequently affected by flash droughts. The average number of events



occurred for the whole of the UK is around 28 events during the growing-season for the
period 1969-2021, although there are some relevant differences between regions. In
general, the Transition (TRAN) and North-West (NW) regions were affected more
frequently compared to South-East (SE) region. Also, SE region shows the higher
variability due to the contrasts observed in the average number of flash droughts reported
across the region.

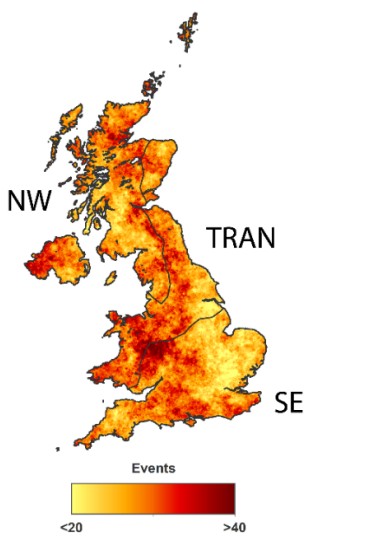

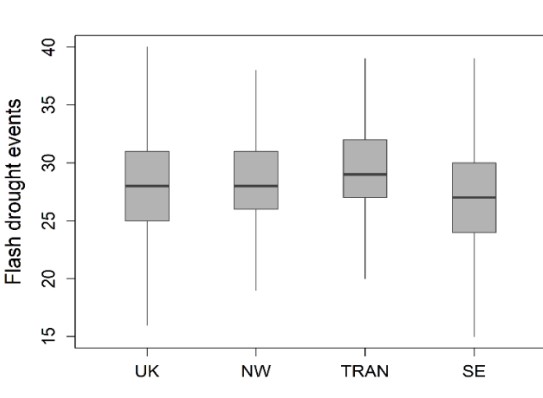

**Figure 2.** Spatial distribution of the total number of flash droughts during the growing-
season (from March to September) in United Kingdom for the period 1969-2021.
Figure 3 shows the seasonal frequencies of flash droughts (events/pixel) in the
UK for each season over the period 1969-2021. The seasonal series show a high
interannual variability, highlighting the period around the late 1980s and early 1990s in
which UK was frequently affected by flash droughts. Overall, non-significant trends are
observed, with negative and non-significant trends in winter, summer, and autumn. In
contrast, there is a positive and significant increase in the number of flash droughts in
spring. At the regional scale, seasonal series also reflect a high variability and generally
non-significant trends (Figure A2). In winter, the Transition (TRAN) and South-East (SE)
regions show no relevant changes in the frequency of flash droughts, while a slight and
non-significant decrease in the number of events is reported in the North-West (NW)
region. On the contrary, positive trends are observed in all regions in spring, although
these trends are only significant in NW and TRAN regions. In summer, there are



important differences between the NW and TRAN region, with a negative and even
significant trend in the case of TRAN region, and positive and non-significant trend in
the SE region. The autumn series show negative and non-significant trends in all regions,
but especially in SW and TRAN regions as a result of the high occurrence of flash
droughts in the early decades of the series.

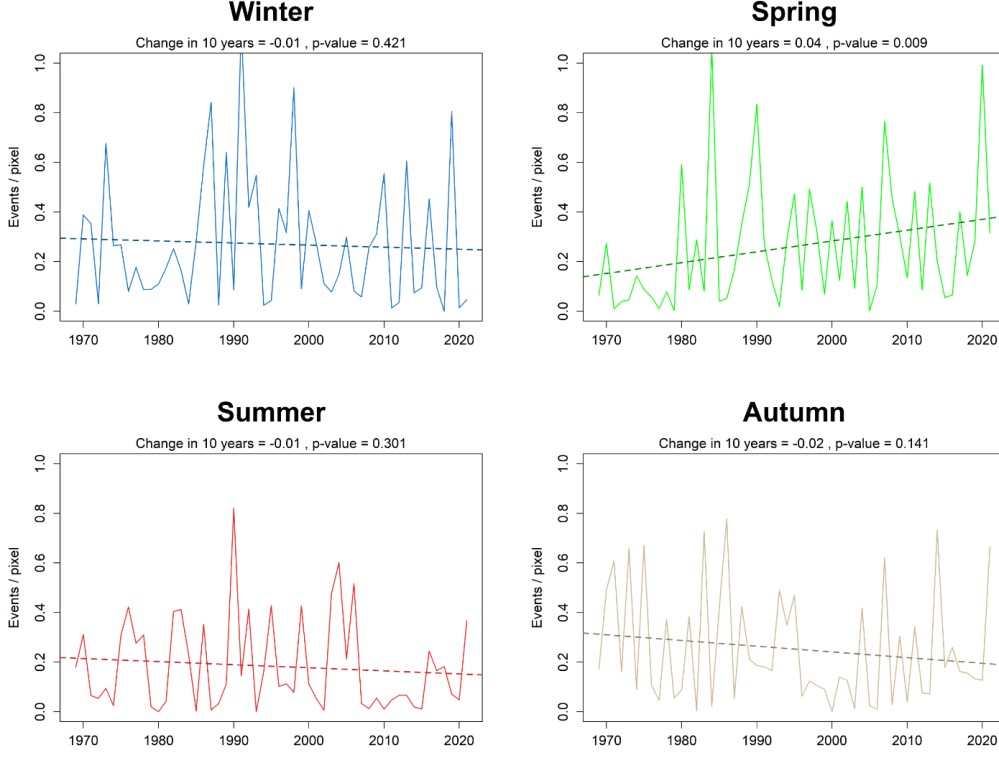


**Figure 3.** Seasonal evolution of the number of flash droughts (events/pixel) in United
Kingdom for the period 1969-2021.
The spatial distribution of the seasonal trends of flash droughts for the period
1969-2021 is depicted in the Figure 4. In general, there are important spatial and seasonal
differences in the trends observed. Non-significant trends over most of the UK are record
in winter months, and only a few small areas in the north show a significant trend. In
spring, there is a clear dominance of positive trends, which are significant in many areas
across the UK. Negative and non-significant trends predominate in summer months,
except for the southeastern UK, where positive and generally non-significant trends are



noted. In autumn, negative and non-significant trends are also record over most of the
UK, except for some small areas in northern region.

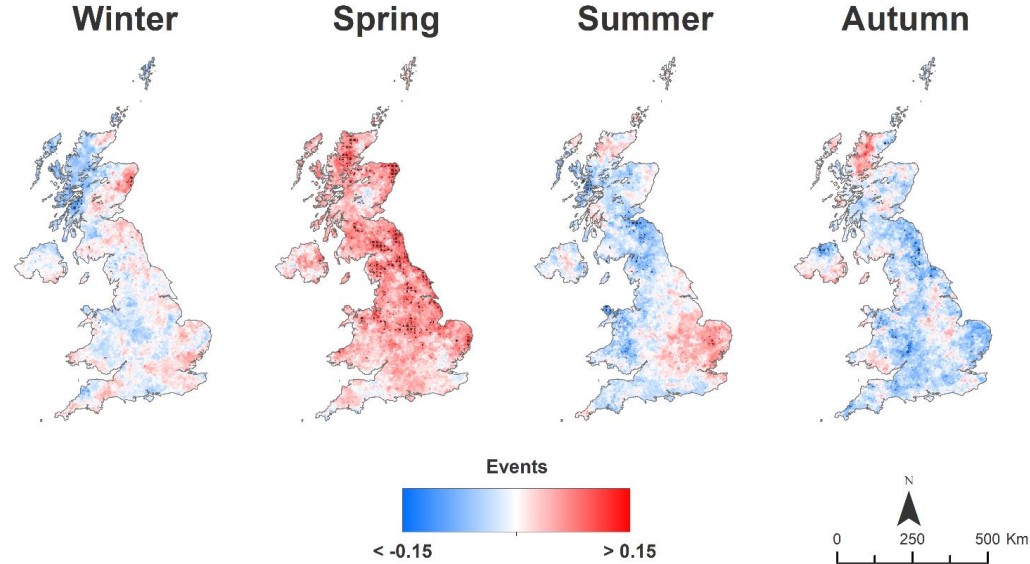


**Figure 4.** Spatial distribution of the seasonal magnitudes of change per decade in flash
drought events in the United Kingdom for the period 1969-2021. Dotted areas represent
those areas in which significant trends were reported.

During the growing-season, non-significant trends are noted for the whole of the

UK, although there are important spatial differences in the magnitude and sign of the
trends (Figure 5). Positive trends were generally reported in eastern and northern regions,
observing significant increases in some areas around southeastern and northern UK. By
contrast, negative and non-significant trends predominate over the west of the UK. There
are also important differences in the frequency of events identified during the growing-
season in each region, although non-significant increases are observed. Highlight period
by a high occurrence of flash droughts in 1980-1990 over NW and TRAN, and in 2000-
2010 over TRAN and SW regions.





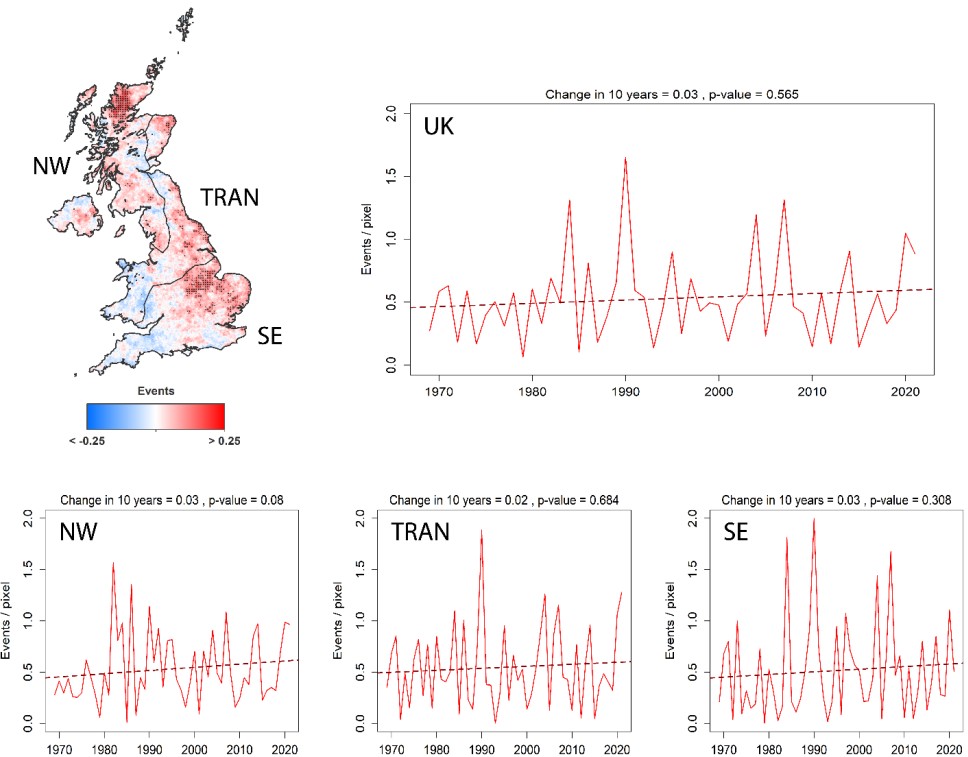

**Figure 5.** Magnitude of change per decade in the flash drought frequencies (events/pixel) observed during the growing-season (from March to September) over the United Kingdom for the period 1969-2021. Dotted areas represent those areas in which significant trends were reported.

### 3.2 Flash drought response to precipitation and AED

Figure 6 shows the seasonal spatial distribution of the average contribution of the atmospheric evaporative demand (AED) to flash drought development in the UK for the period 1969-2021. As expected, the contribution of the AED to flash drought development shows large spatial and seasonal contrasts as a result of the large climatic variability of UK (Figure A3). In general, the average AED contribution exhibits a strong spatial coherence with the average precipitation at seasonal scale (Figure A3a). In winter, when the precipitation is very high and AED rarely exceeds 50mm, the average AED contribution is close to zero over most of the UK except for some areas of the east. The maximum values of the AED contribution are found in spring months, with large areas over central, eastern, and especially southeastern UK exceeding 15%. In these areas, the average precipitation reaches its seasonal minimum, while the AED increases notably





379 compared to the winter months. The AED contribution in summer also depicts average

380 values around 15% in a few areas of the south, where the average precipitation is lower

381 and the average AED reaches its maximum values (Figure A3b), but in general most of

382 the UK shows a low average AED contribution to flash drought development. In autumn,

383 with the increase in precipitation and the decline in AED, most of the UK shows average

384 AED contribution values close to zero and only some areas of the east record higher

385 average values (5-10%).

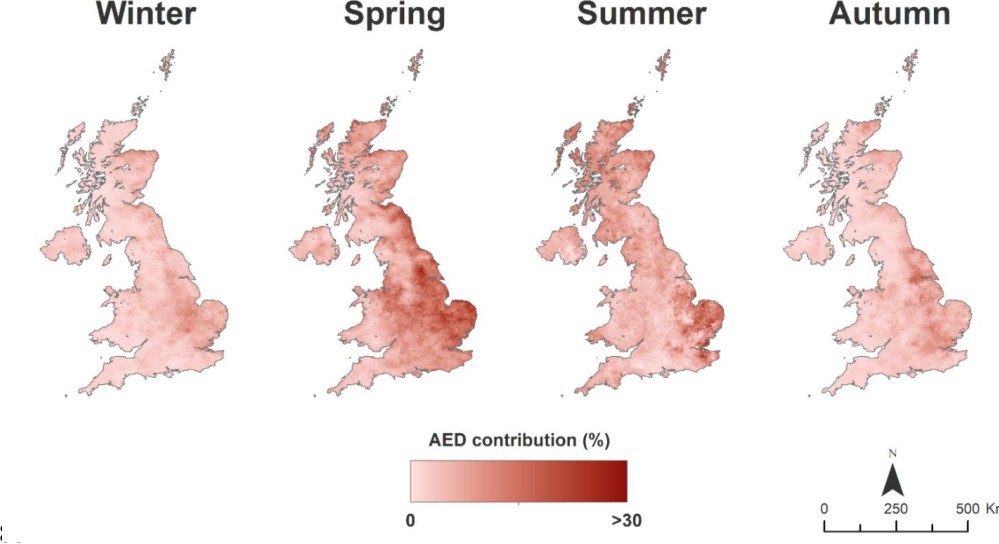

**Figure 6.** Seasonal spatial distribution of the average contribution of AED to flash
drought development in United Kingdom for the period 1969-2021.

389   The evolution of the average AED contribution to flash drought development

390 also exhibits important interannual variations in each season over the period 1969-2021

391 (Figure 7). There is a significant increase in AED contribution in spring, which is

392 particularly notable since the early 1990s. No relevant changes are noted in winter and

393 autumn, while there is a slight and non-significant increase in the AED contribution in

394 summer. In general, the changes reported in the average AED contribution to flash

395 drought shows a consistent relationship with the trends observed in the average rainfall

396 and AED at seasonal scale (Figure A4). Thus, spring, the only season with a significant

397 increase in AED, is also the only season that does not show an increase in rainfall, which

398 additionally concurred with a significant increase in AED.





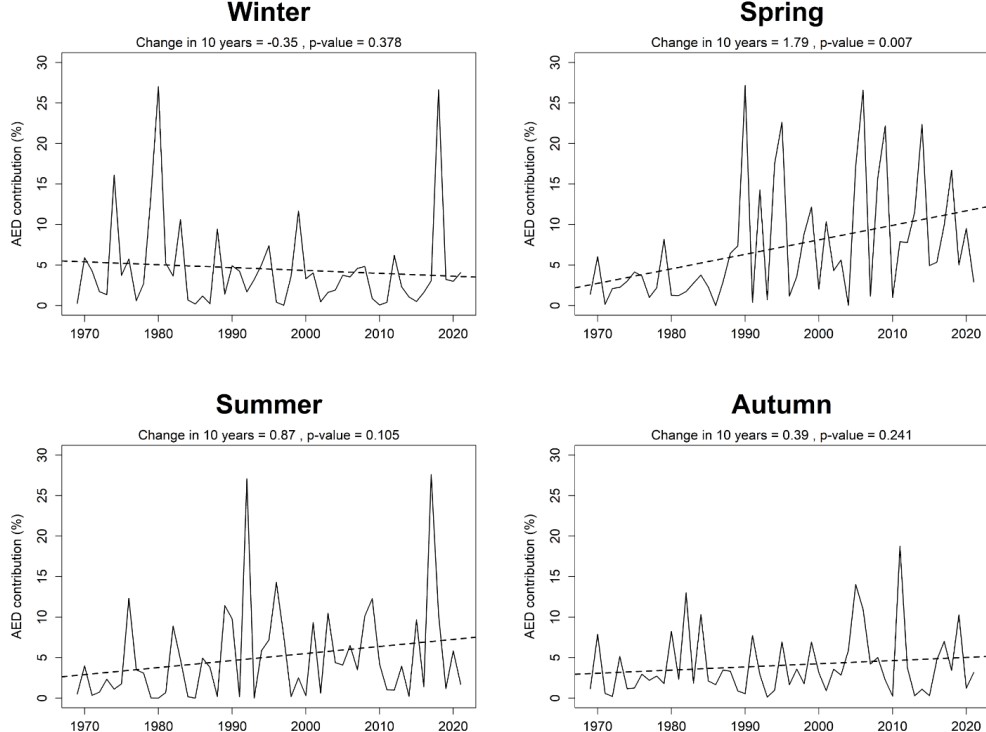

**Figure 7.** Seasonal evolution of the average contribution of AED to flash drought development in United Kingdom for the period 1969-2021.

At regional scale, some relevant differences in the evolution of the AED contribution are noted (Figure A5). A decrease in AED contribution is recorded in TRAN and SE region in winter, although only the SE region exhibits a significant trend. By contrast, all regions show an increase in AED contribution in spring, which is significant in NW and TRAN regions. In summer, a general increase in AED contribution is recorded, but this increase only is significant in SE region. In autumn, a significant decrease in AED contribution is recorded in NW region, while regions TRAN and SE show non-significant increases. In general, there is also a clear regional relationship between the evolution of AED contribution and precipitation and AED patterns in each region (Figure A6 and S7).

**3.3 Atmospheric and oceanic conditions during flash drought development**





Figure 8 shows the seasonal composites of 500 hPa geopotential height (Z500)

and sea level pressure (SLP) anomalies during the development of the top-10 flash
droughts recorded in each season for the period 1969-2021. Overall, notable positive
Z500 anomalies are recorded during flash droughts development over the UK and western
Europe in all seasons, exceeding 50m in summer and spring, or even 100m in winter and
autumn. Similarly, high SLP anomalies are recorded during flash droughts development
in all seasons, although there are some seasonal variations. The highest anomalies in SLP
are recorded in winter, with values higher than 10 hPa around UK. Notable anomalies in
SLP are also noted in spring and autumn, exceeding 6 hPa. In summer, the positive
anomalies reach the lowest values (2-4 hPa).

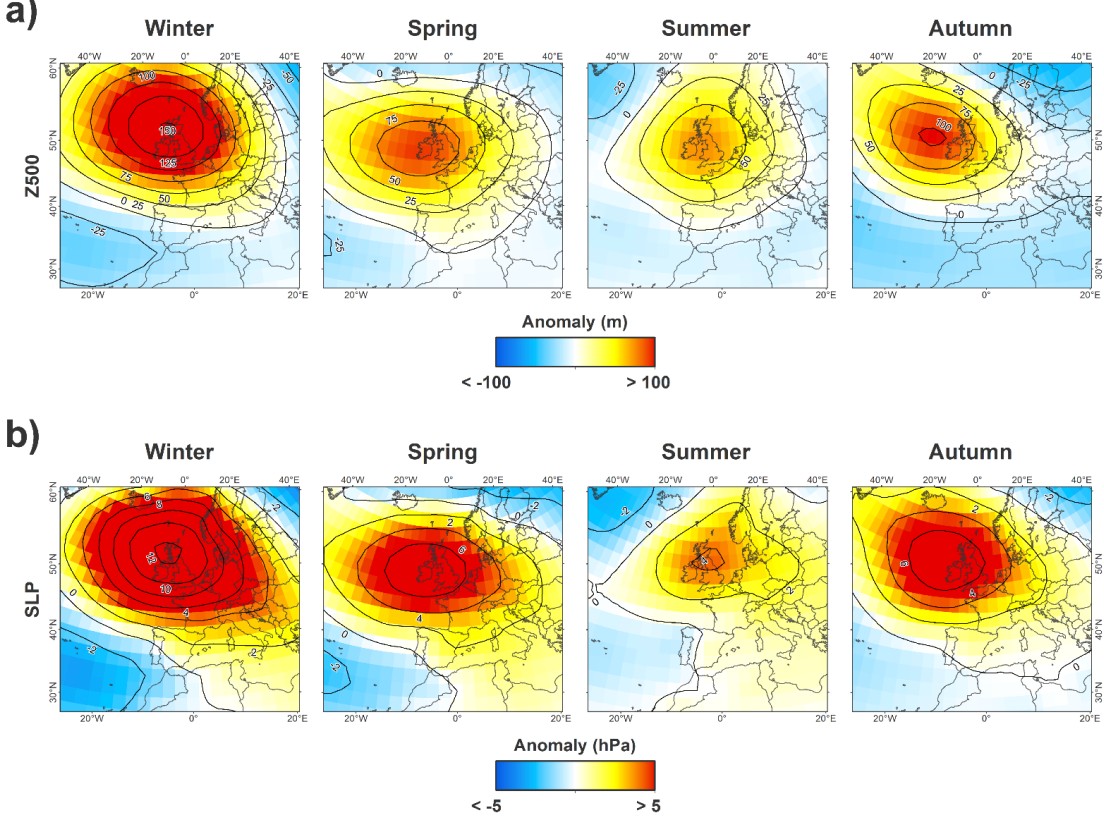

**Figure 8.** Seasonal composites of **(a)** Z500 and **(b)** SLP anomalies during the
development of the top-10 flash droughts of each season over the United Kingdom for
the period 1969-2021.



The average anomalies in North Atlantic Oscillation index (NAOi) during the
development of the top-10 flash droughts of each season are presented in Figure 9.
Important seasonal differences were noted in NAO phase during the development of flash
droughts, with a marked contrast between winter-autumn and summer-spring months. In
winter and autumn, remarkable and negative anomalies in NAOi are recorded, with
average values around -1, but in some cases are less than -2. By contrast, positive and
moderate NAOi anomalies are dominant during the develop of the flash droughts ocurred
in spring and summer months.

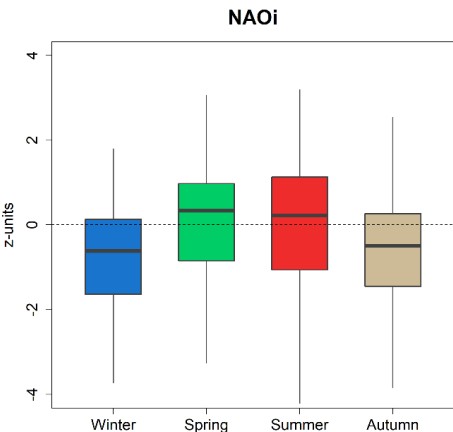


**Figure 9.** Seasonal North Atlantic Oscillation index (NAOi) values during the
development of the top-10 flash droughts of each season over the United Kingdom for
the period 1969-2021.

Finally, the seasonal anomalies in sea surface temperature (SST) were examined
during the development of the top-10 flash droughts recorded in each season for the
period 1982-2021 (Figure 10). Positive SST anomalies are generally recorded during the
development of the flash drought in spring and summer over Atlantic Ocean around the
UK and western Europe coast, with anomalies that generally exceed 1ºC in summer
months. By contrast, we found a higher spatial variability in SST during winter and
autumn, with both positive and negative anomalies recorded during the development of
flash drought in these seasons over Atlantic Ocean around UK. Positive and remarkable
anomalies were also observer over some areas of the Arctic Ocean in all seasons, which
exceed 1ºC.





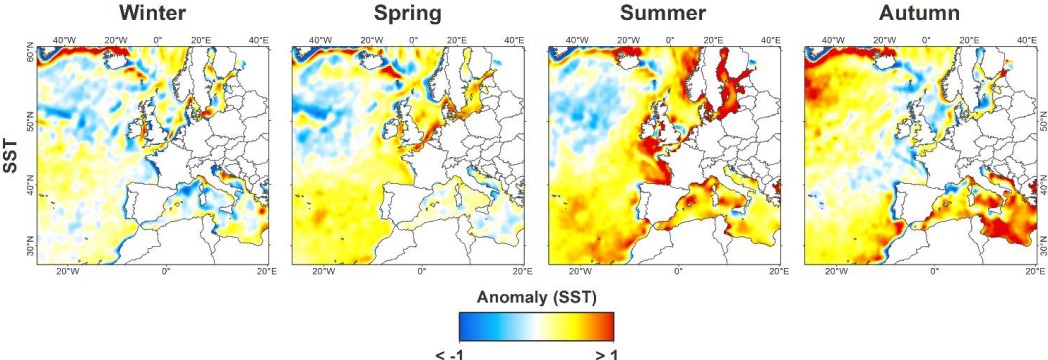

4: _

**Figure 10.** Seasonal anomalies (ºC) in sea surface temperature (SST) during the development of the top-10 flash droughts of each season over the United Kingdom for the period 1982-2021.

## 4. Discussion

### 4.1 Characteristics and trends of flash droughts in UK

This study analysed the occurrence of flash droughts in the UK over a long-term period. The results evidence that flash drought is characterised by a high variability, with important regional and seasonal differences. Droughts in UK exhibits a great spatiotemporal variability (Tanguy et al., 2021) and, naturally, this complexity also extends to flash drought patterns. However, the patterns of these rapid-onset droughts occurred at short times scales vary notably from those found by previous studies focused on long-term droughts (Burke and Brown, 2010; Dobson et al., 2020; Rahiz and New, 2012). Our finding shows that both the wetter regions of the North-West and the drier areas of the South-East were affected by flash drought in all seasons over the last five decades. Overall, the highest frequency of flash drought is reported in Wales and Northern Ireland, while the southeastern regions reported the lowest number of events. The high number of events recorded in some humid regions of the central and northern UK could be a response to the frequent occurrence of short dry periods compared to the southeastern regions, where rainfall is notably lower as well as less variable, so these rapid dry spells may be less frequent but more relevant in terms of impacts. For example, Tanguy et al. (2021) found that northwestern regions tend to be more frequently affected by short-term droughts, while the southeastern regions are affected by droughts less frequently but with greater severity. In late autumn and winter, it is expected that flash droughts have little environmental impact as deficits built up during short dry periods are quickly replenished by wet periods, although these dry spells may still be relevant from a hydrological point





of view given the quick response (~1-month) of UK catchments to rainfall scarcity,
especially in the north (Barker et al., 2016). Conversely, flash droughts occurring in
spring, summer, and early autumn (i.e. growing-season), which affect central and western
UK more frequently, are expected to have important environmental and agricultural
implications. During this period vegetation demands more water and precipitation deficits
associated with droughts are often accompanied by increased temperatures leading to
vegetation stress (Pribyl, 2020), with attendant environmental and agricultural impacts,
as apparent during recent summer half-year droughts (Barker et al., 2024; Turner et al.,

2021).

In general, there are no compelling major increases in flash drought frequencies

for the period 1969-2021. Previous studies focused on long-term drought (e.g. 3-, 6- and
12- months times scales) also reported few changes in drought occurrence over most of
the UK (Tanguy et al., 2021; Vicente-Serrano et al., 2021). Nevertheless, we found a
notable and significant increase in the number of flash droughts recorded in spring.
Recent studies based on soil moisture data from reanalysis suggest an increase in flash
drought frequency at European scale associated with the rise of evaporative demand in
the last few years (Shah et al., 2022). In this case, we noted some parallels between the
trends in flash droughts and the recent evolution of rainfall and AED over UK at seasonal
scale (see Figure A4). Thus, the only season in which precipitation has not increased and
AED has raised significantly (i.e., spring), is the only one that shows a general increase
in flash drought frequency. On the contrary, the seasons in which the average precipitation
has increased show generally negative and non-significant trends. Therefore, there is a
seasonal consistency between flash drought frequencies and the spatiotemporal patterns
noted in rainfall and AED over UK. During the growing-season, when the impacts of this
kind of events are expected to be greater, we observed significant increases in the eastern
regions due to the increase in the number of events observed in spring and summer over
these areas, although there is no clear trend for the whole of the UK as well as for each
of the regions considered.

## 4.2 Meteorological drivers underlying flash droughts


Flash droughts in the UK are strongly driven by precipitation variability,

particularly in winter and autumn. In these cold and wet months in which AED is very
low (Mayes and Wheeler, 2013), drought triggering depends almost exclusively on the
occurrence of deficits in rainfall and AED is irrelevant with a few exceptions. The results





evidenced that AED is only relevant in the drier regions of the southeast in spring and
summer, when rising temperature (e.g. associated with heat wave episodes) combined
with precipitation deficit can exacerbate pressure on water resources, amplifying drought
impacts (Turner et al., 2021). By contrasts, in humid regions such as northern UK, AED
has a minor role in triggering droughts. In these regions characterised by energy-limited
conditions, under normal (wet) conditions, an increase in AED would have no impacts
(Vicente-Serrano et al., 2020). Thus, it is expected that AED is only relevant in driving
drought conditions during very dry periods as rainfall is a key factor determining the
effect of AED on drought (Tomas-Burguera et al., 2020). Indeed, there is a clear spatial
relationship between mean precipitation and the AED contribution to flash drought,
which shows the same northwest-southeast gradient observed in rainfall distribution.
Although rainfall is the primary factor controlling flash drought variability in the
UK, we found that the role of AED is becoming more relevant in triggering summer and
spring flash droughts. This is especially evidenced in spring, when a significant increase
in AED was noted, but also in southeastern region in summer. Curiously, the maximum
percentages of AED contribution to flash drought development were generally found in
spring rather than in summer. This pattern may be explained by the notable increase in
AED contribution in spring since late 1980s associated with the general rise of AED in
this season (Blyth et al., 2019; Robinson et al., 2017), but also by the anomalous higher-
than-average precipitation recorded during summer (Kendon et al., 2022) compared to
spring over recent few years. In other words, spring was the driest season in UK over the
last five decades. The trends observed in AED contribution could be relevant to
understand the recent trends observed in flash droughts occurrence in summer and,
particularly, in spring. We found that those regions and seasons, in which AED
contribution increased, generally show positive trends in flash drought frequency.
Previous studies have linked the increase in the frequency and severity of flash droughts
in some regions of the world to the growing relevance of AED as a driver of drought
conditions under global warming (Mishra et al., 2021; Noguera et al., 2022; Yuan et al.,

2018, 2019).

### 4.3 Atmospheric and oceanic conditions involved in flash drought development

Flash droughts development is strongly associated to the presence of high-
pressure systems over the UK. Remarkable anomalies in SLP and Z500 were noted during





the development of flash droughts in all seasons, but particularly in winter. The patterns
observed typically respond to the northward displacement of the Azores High, resulting
in blocking situations that prevent the arrival of humid air masses and, consequently,
inhibiting precipitation (Richardson et al., 2018). In winter and autumn, the location of
the pressure fields corresponds to the typical patterns of the negative phase of the NAO.
Thus, the development of flash droughts in autumn and particularly in winter, is
commonly associated with strong negative anomalies in NAOi. Numerous studies have
demonstrated the relationship between the negative phase of the NAO and the absence of
precipitation during these seasons (Fowler and Kilsby, 2002; Murphy and Washington,
2001; West et al., 2021b), particularly in northwestern regions (West et al., 2019). In
addition, the negative phase of the NAO in winter usually coincides with cold periods
(Hall and Hanna, 2018), which would reinforce the negligible role of the AED compared
to that of rainfall during these months. On the contrary, positive anomalies in NAOi are
generally recorded in spring and summer, although these anomalies are highly variable.
During these months, there is not a strong relationship between precipitation variability
and NAO phase (West et al., 2021b), which would explain why the anomalies recorded
during these months are generally more variable. NAO is the main large-scale
atmospheric circulation pattern that control precipitation variability (West et al., 2021a),
and its links with drought occurrence is well-know (West et al., 2022). The anomalies
observed during the previous weeks to flash drought onset confirm that flash drought
development is also closely connected with NAO phase, especially in winter.

Flash droughts usually develop during period of positive SST over the Atlantic

Ocean around UK and western Europe coast in spring and summer, while no clear patterns
in SST anomalies are recorded in winter and autumn flash droughts. The influences of
SST on drought are quite complex considering the strong oceanic-atmospheric
interactions and its crucial role modulating large-scale atmospheric circulation patterns
(Robertson et al., 2000). Several studies showed how SST anomalies over the Atlantic
Ocean can have an important role driving precipitation and, consequently, drought
variability over Europe at long-term (Ionita et al., 2015; Rimbu et al., 2001). Recent
studies also noted that SST anomalies can play certain role driving drought events
developing at short-term as flash droughts (Ma et al., 2024). In the case UK, SST patterns
over the Atlantic Ocean are very important in promoting drought occurrence given their
influence on atmospheric circulation, including the NAO (Kingston et al., 2013; Svensson



and Hannaford, 2019). Here, we found some similarities with the patterns observed for other studies that showed a connection between drought occurrence in UK and periods characterised by positive SST anomalies in eastern Atlantic Ocean and the Artic Ocean prior to the onset of spring and summer drought (Kingston et al., 2013; McCarthy et al., 2019). This seems to suggest that these anomalies may have some relevance in favouring the development of flash drought events, although this issue requires further research.

### 4.4 Limitations and future work

Despite the consistency of the results with the meteorological observations as well as the ocean-atmospheric conditions, there are some issues that should be carefully considered in interpreting our findings. Firstly, adopting an approach for flash drought identification based exclusively on meteorological data does not provide a measure of drought impacts. In addition to meteorological data, a comprehensive assessment of drought conditions would ideally require the use of different source of data, including; data on vegetation activity, soil moisture and streamflow variability, or crop yield, among others (Otkin et al., 2022). Some of these datasets have constraints (e.g. relatively short records) so we focused our study meteorological data that enabled us to carry out our study at long-term. Future work could link flash drought occurrence, as reported here, with hydrological drought responses and agricultural or environmental impacts. Moreover, applying a method focused only on the rate of intensification of the development phase to identify flash drought, it is expected that in some cases the strong deficits occurring in the short-term could be quickly replaced by wet periods and not have a great relevance in terms of impacts, especially if the development of the event was preceded by humid conditions. This issue is more likely to occur in late autumn and winter, when wet and cold conditions are dominant and vegetation activity is lower.

Another important point that should be considered is related to the complex dynamics of precipitation in UK (Hulme and Barrow, 1997; Mayes and Wheeler, 1997), which is characterised by large variations. Given the great variability of precipitation in UK, the period selected for the analysis had important implications on the trends observed. This is especially crucial in summer season when a high interdecadal variability is observed. For example, given the occurrence of unusual wet summers since 2007 (Kendon et al., 2022), positive trends in precipitation are recorded over the last decades, as well as increases in stream flows (Hannaford, 2015). By contrast, other studies focussing on very long records (i.e. period 1776-2002) found a decrease in summer





precipitation over England and Wales (Mills, 2005). Therefore, although summer got
wetter if we consider the last few decades, these trends are strongly determined by the
period selected and could vary notably when considering longer records.
Future work should focus on addressing whether the observed trends are simply
due to natural climate variability, or whether these increases could be attributed to
anthropogenic forcing contributing to rising temperatures and the relevance of the AED
on flash drought development. In this way, large ensembles could be considered in the
future to examine possible trends according to natural variability (e,g. Deser and Phillips,
2023). Furthermore, it would be necessary to analyse future projections of these trends
under different greenhouse emission scenarios to disentangle the possible effect of
climate change on the occurrence of flash droughts in the UK. Another key issue that
should be analysed in future studies is the response of the different systems affected by
drought, as well as unravelling how flash drought conditions propagate through these
systems in UK. The response of crops, natural vegetation, soil moisture and river flows
should be analysed to unravel how the meteorological anomalies identified in this study
translate in terms of impact, given that the response of the different affected systems is
expected to vary considerably over time and space. There are increasing efforts to
establish databases of the environmental and social impacts of drought, which could also
be linked to flash drought occurrence (e.g. building on previous approaches applied for
droughts more generally, e.g. Bachmair et al. 2015, Parsons et al. 2019).

## 5. Conclusion

In this research, we present for the first time a climatology of flash droughts in
UK, providing a detailed characterisation of their spatial and temporal patterns. Likewise,
we analysed the trends in the seasonal occurrence of flash droughts over the last five
decades. We also show the role played by AED on flash drought triggering, as well as its
evolution under the currently process of global warming. Finally, we analysed the
atmospheric and oceanic conditions recorded during flash droughts development, and
their possible connections with large-scale atmospheric patterns such as NAO. The main
conclusions from this study are as follows:
- Flash drought occurrence in UK is characterised by a high spatial and seasonal
variability, affecting both the wetter regions of the North-West and the drier
regions of the South-East.





• There is a notable and significant increase of flash droughts in spring, but non-
significant trends (positive/negative) noted in winter, summer and autumn.
• Flash droughts in UK are mainly driven by rainfall variability, while the AED has
a minor role triggering flash drought occurrence. In spring, there is a significant
increase in AED contribution, which could explain the positive and significant
trends reported in the number of events in this season.
• Positive and remarkable anomalies in SLP and Z500 were noted during the flash
droughts development in all seasons. These anomalies are associated with the
presence of high-pressure systems around UK, which prevent the arrival of humid
air masses and, consequently, inhibit precipitation.
• North Atlantic Oscillation (NAO) strongly controls flash droughts occurrence
over the UK, particularly in winter and autumn months.
• Positive anomalies in sea surface temperatures (SST) were seen over the Atlantic
Ocean around UK during flash drought development in spring and summer, while
mixed anomalies were observed in winter and autumn.














**Appendix A**

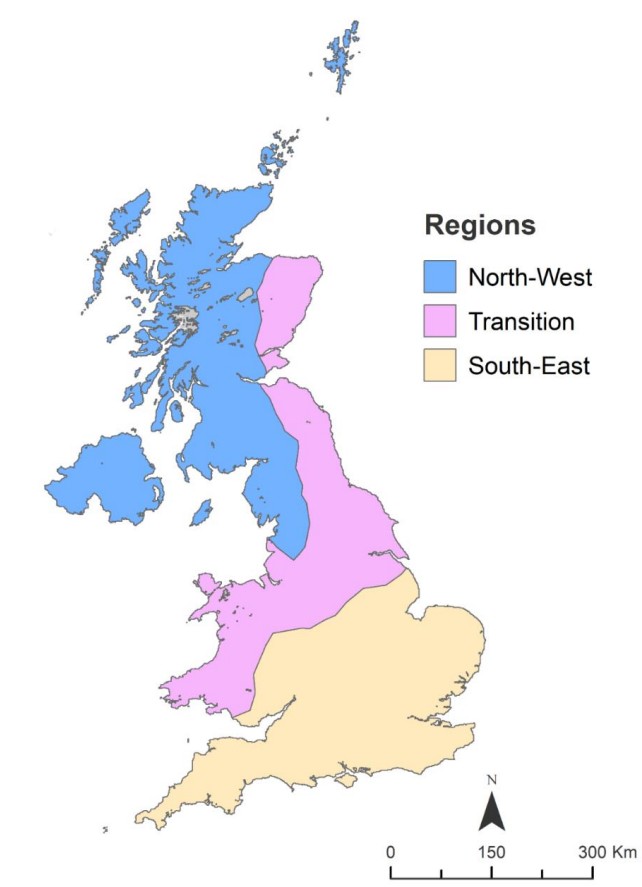


**Figure A1.** Regional delimitation based on Maliko et al. (2021).






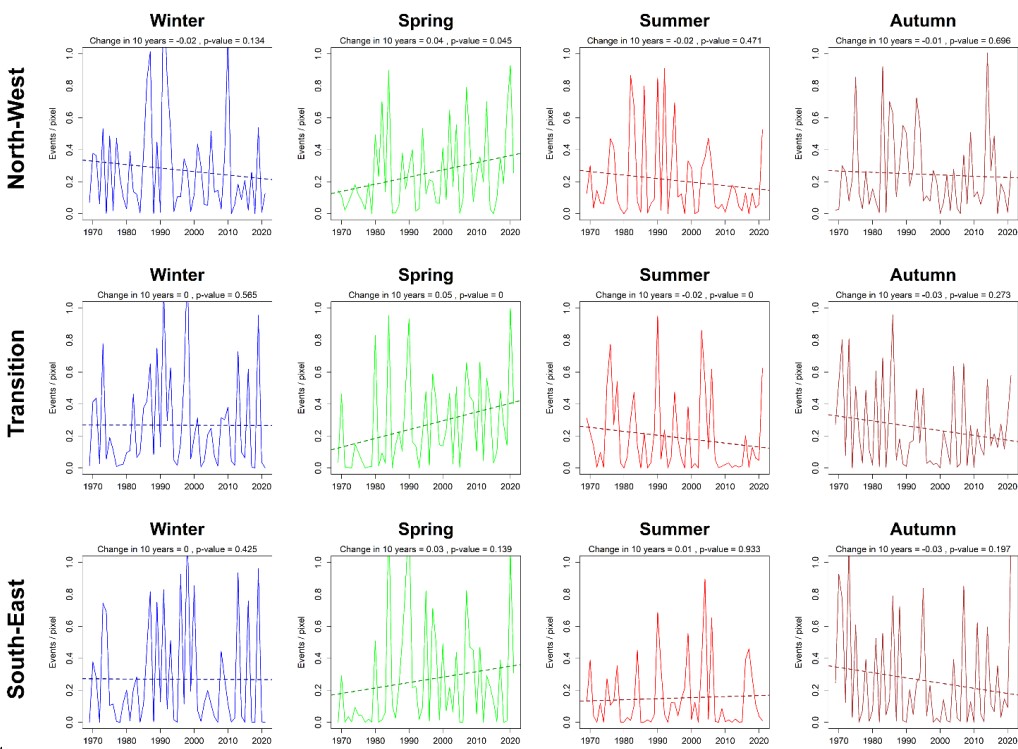

6′. .

**Figure A2.** Seasonal evolution of the number of flash droughts (events/pixel) in United
Kingdom for the period 1969-2021 by regions.



a)

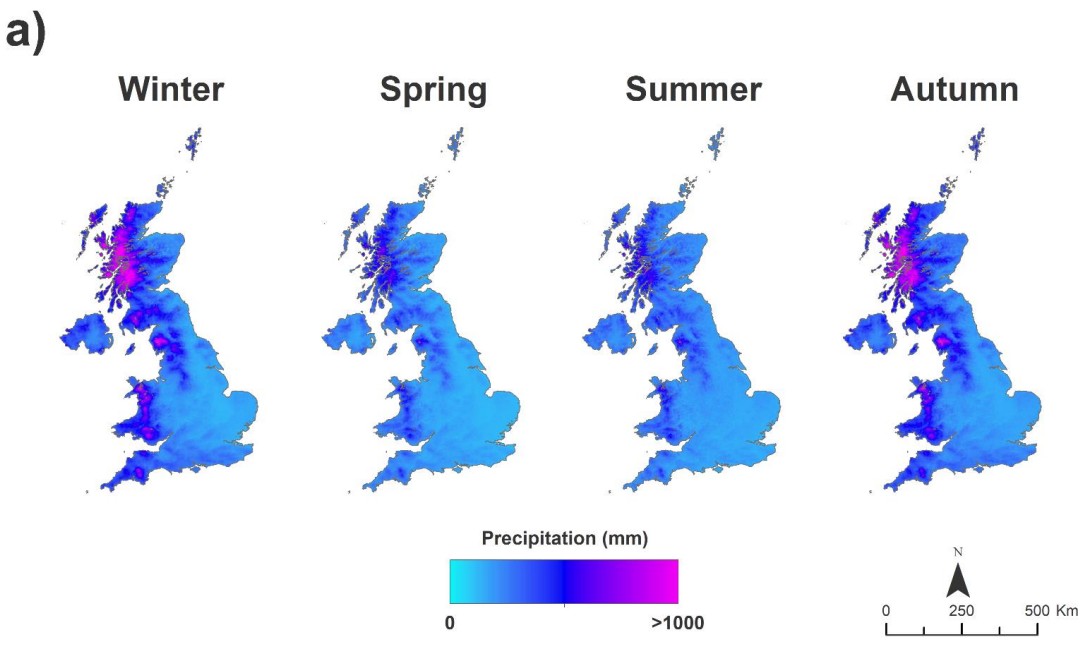

b)

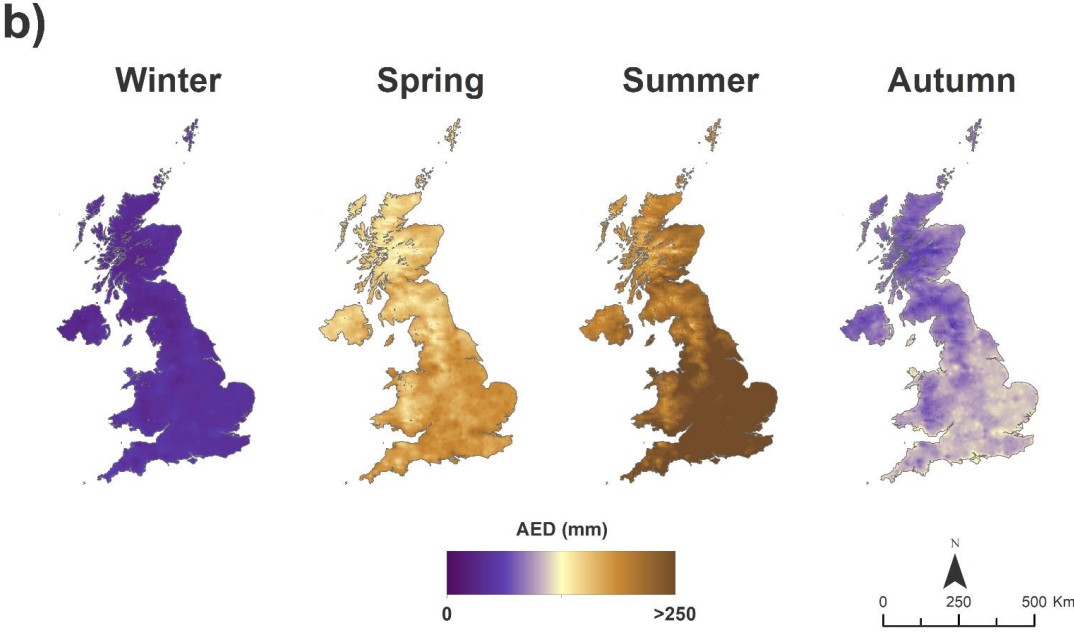

**Figure A3.** Seasonal spatial distribution of the average **(a)** precipitation and **(b)** AED in United Kingdom over the period 1969-2021.



a)

b)

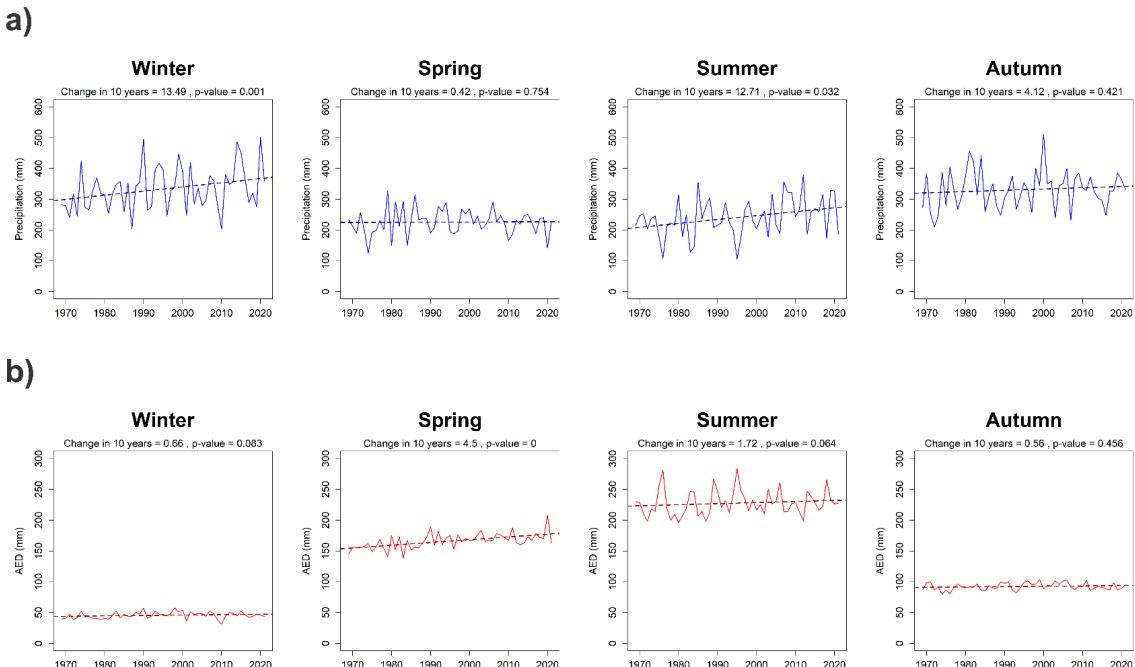

**Figure A4.** Seasonal evolution of the average **(a)** precipitation and **(b)** AED in United Kingdom for the period 1969-2021.





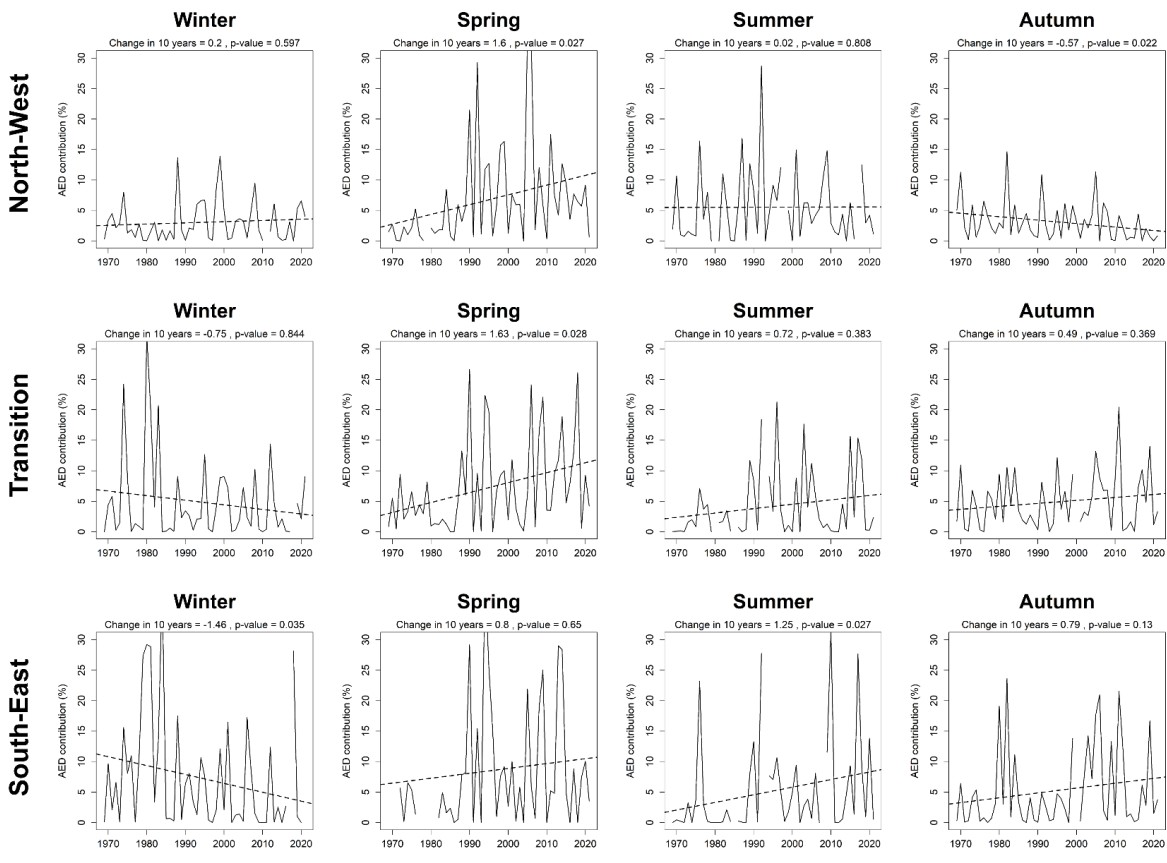

**Figure A5.** Seasonal evolution of the average contribution of AED to flash drought development in United Kingdom for the period 1969-2021 by regions.



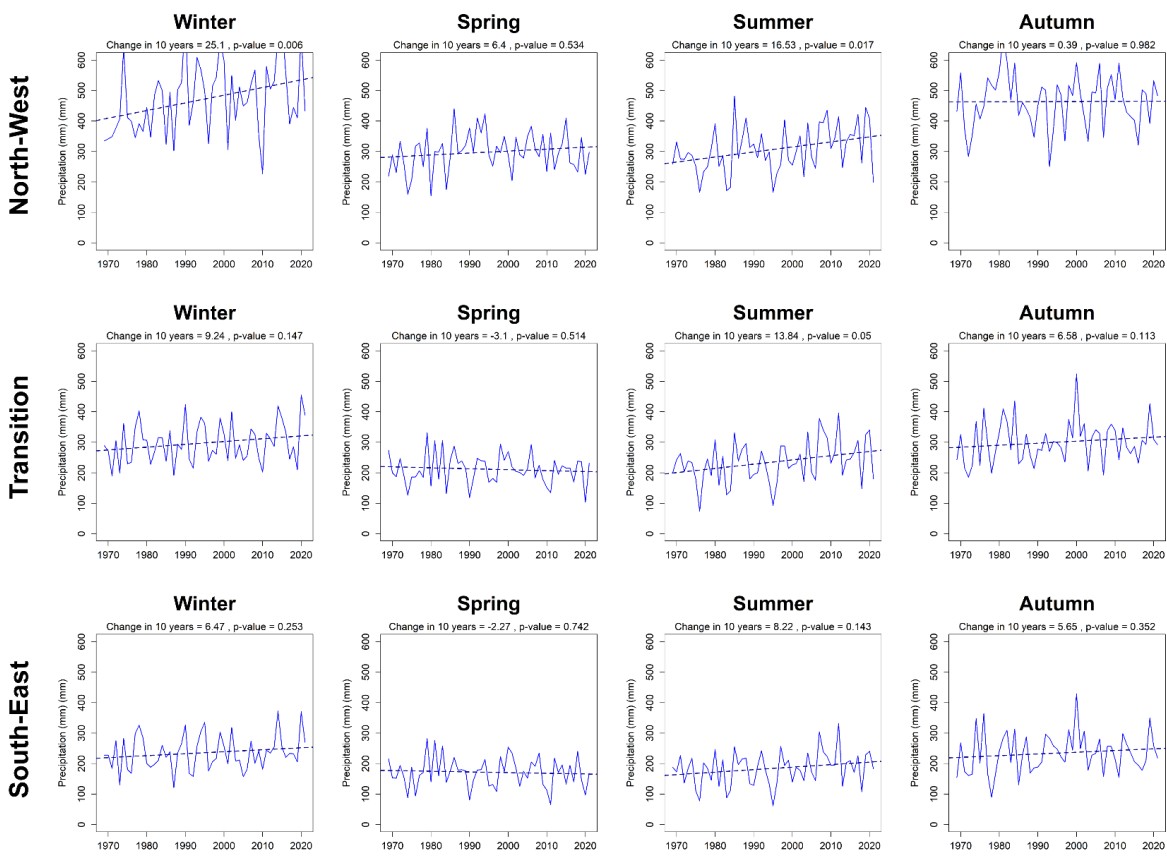

**Figure A6.** Seasonal evolution of the average precipitation in United Kingdom for the
period 1969-2021 by regions.













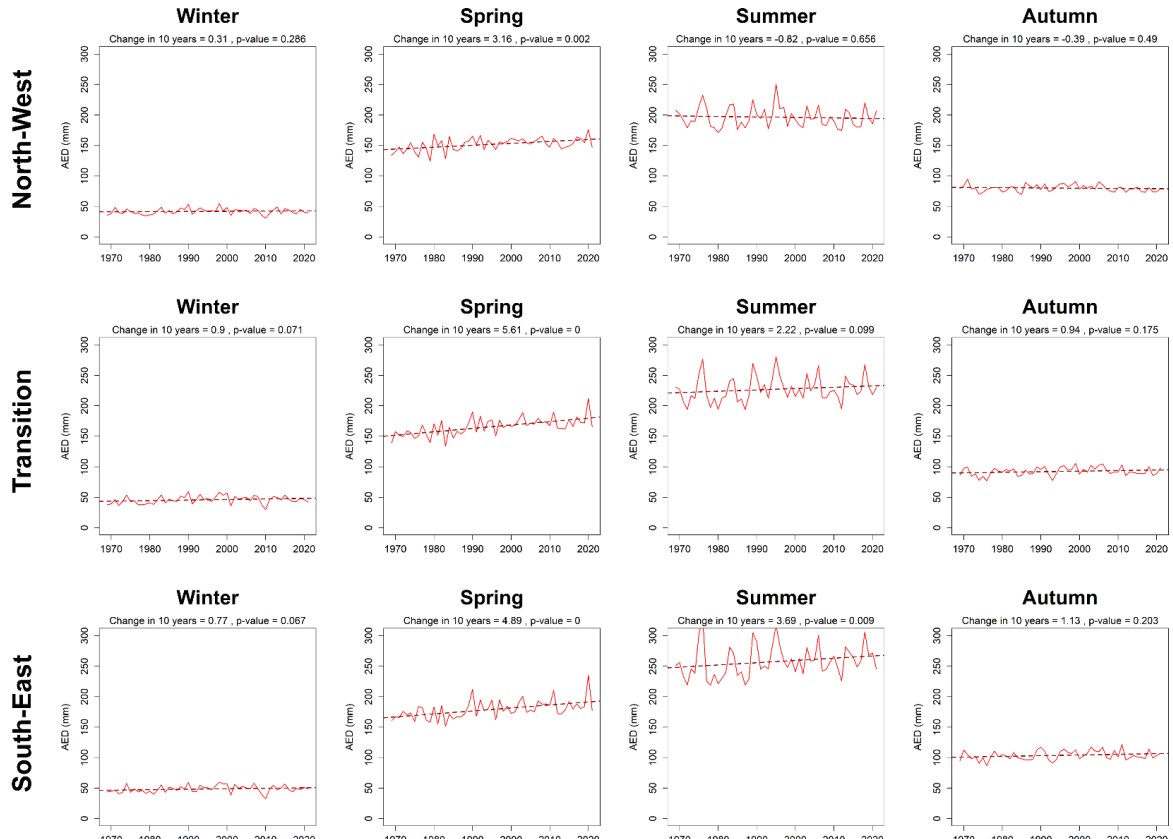

**Figure A7.** Seasonal evolution of the average atmospheric evaporative demand (AED) in United Kingdom for the period 1969-2021 by regions.

## Author contribution

All authors contributed to the conceptualisation and design of the research, as well as to the preparation and revision of the manuscript. IN conducted the data processing, analysis and visualisation.

## Competing interests

The authors declared that there are no competing interests.

## Acknowledgements

This study was funded by the Natural Environment Research Council under the HydroJULES Programme (NE/S017380/1).



750

**Data availability**

All information used in this study is open access. To calculate SPEI, we employed daily
precipitation and AED data. Precipitation data was obtained from Met Office Hadley
Centre for Climate Science and Services, which is available at
https://catalogue.ceda.ac.uk/uuid/4dc8450d889a491ebb20e724debe2dfb. While AED
data was obtained from Environmental Information Data Centre (EIDC), which is
available at https://catalogue.ceh.ac.uk/documents/beb62085-ba81-480c-9ed0-
2d31c27ff196. To analysed the atmospheric and oceanic conditions during flash drought
development, we employed daily sea level pressure (SLP), 500 hPa geopotential height
(Z500) and sea surface temperature (SST) from the National Centers for Environmental
Prediction (NCEP)–National Center for Atmospheric Research (NCAR), which is
available at https://psl.noaa.gov/data/.

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

time scales of climatological drought: An evaluation of the Standardized Precipitation
Index in a mountainous Mediterranean basin, Hydrol Earth Syst Sci, 9, 523–533,
https://doi.org/10.5194/hess-9-523-2005, 2005.
Vicente-Serrano, S. M., Beguería, S., and López-Moreno, J. I.: A multiscalar drought
index sensitive to global warming: The standardized precipitation evapotranspiration
index, J Clim, 23, 1696–1718, https://doi.org/10.1175/2009JCLI2909.1, 2010.
Vicente-Serrano, S. M., Gouveia, C., Camarero, J. J., Beguería, S., Trigo, R., Lopez-
Moreno, J. I., Azorín-Molina, C., Pasho, E., Lorenzo-Lacruz, J., Revuelto, J., Moran-
Tejeda, E., and Sanchez-Lorenzo, A.: Response of vegetation to drought time-scales
across global land biomes, Proceedings of the National Academy of Sciences, 110, 52–
57, https://doi.org/10.1073/pnas.1207068110, 2013.
Vicente-Serrano, S. M., Camarero, J. J., and Azorín-Molina, C.: Diverse responses of
forest growth to drought time-scales in the Northern Hemisphere, Global Ecology and
Biogeography, 23, 1019–1030, https://doi.org/10.1111/geb.12183, 2014.
Vicente-Serrano, S. M., McVicar, T. R., Miralles, D. G., Yang, Y., and Tomas-
Burguera, M.: Unraveling the influence of atmospheric evaporative demand on drought
and its response to climate change, WIREs Climate Change, 11,
https://doi.org/10.1002/wcc.632, 2020.



Vicente-Serrano, S. M., Domínguez-Castro, F., Murphy, C., Hannaford, J., Reig, F.,
Peña-Angulo, D., Tramblay, Y., Trigo, R. M., Mac Donald, N., Luna, M. Y., Mc
Carthy, M., Van der Schrier, G., Turco, M., Camuffo, D., Noguera, I., García-Herrera,
R., Becherini, F., Della Valle, A., Tomas-Burguera, M., and El Kenawy, A.: Long-term
variability and trends in meteorological droughts in Western Europe (1851–2018),
International Journal of Climatology, 41, E690–E717,
https://doi.org/10.1002/JOC.6719, 2021.
Vicente-Serrano, S. M., Peña-Angulo, D., Beguería, S., Domínguez-Castro, F., Tomás-
Burguera, M., Noguera, I., Gimeno-Sotelo, L., and El Kenawy, A.: Global drought
trends and future projections, Philosophical Transactions of the Royal Society A, 380,
https://doi.org/10.1098/RSTA.2021.0285, 2022.
Walker, D. W., Vergopolan, N., Cavalcante, L., Smith, K. H., Agoungbome, S. M. D.,
Almagro, A., Apurv, T., Dahal, N. M., Hoffmann, D., Singh, V., and Xiang, Z.: Flash
Drought Typologies and Societal Impacts: A Worldwide Review of Occurrence,
Nomenclature, and Experiences of Local Populations, Weather, Climate, and Society,
16, 3–28, https://doi.org/10.1175/WCAS-D-23-0015.1, 2023.
Wang, K., Dickinson, R. E., and Liang, S.: Global Atmospheric Evaporative Demand
over Land from 1973 to 2008, J Clim, 25, 8353–8361, https://doi.org/10.1175/JCLI-D-
1087 11-00492.1, 2012.

Wang, Y. and Yuan, X.: Anthropogenic Speeding Up of South China Flash Droughts as
Exemplified by the 2019 Summer-Autumn Transition Season, Geophys Res Lett, 48,
e2020GL091901, https://doi.org/10.1029/2020GL091901, 2021.
West, H., Quinn, N., and Horswell, M.: Regional rainfall response to the North Atlantic
Oscillation (NAO) across Great Britain, Hydrology Research, 50, 1549–1563,
https://doi.org/10.2166/NH.2019.015, 2019.
West, H., Quinn, N., and Horswell, M.: Monthly rainfall signatures of the north atlantic
oscillation and east atlantic pattern in Great Britain, Atmosphere (Basel), 12,
https://doi.org/10.3390/atmos12111533, 2021a.
West, H., Quinn, N., Horswell, M., Yuan, N., Cheung, K. K. W., and Shukla, R.:
Spatio-Temporal Variability in North Atlantic Oscillation Monthly Rainfall Signatures
in Great Britain, Atmosphere 2021, Vol. 12, Page 763, 12, 763,
https://doi.org/10.3390/ATMOS12060763, 2021b.
West, H., Quinn, N., and Horswell, M.: The Influence of the North Atlantic Oscillation
and East Atlantic Pattern on Drought in British Catchments, Front Environ Sci, 10,
754597, https://doi.org/10.3389/FENVS.2022.754597/BIBTEX, 2022.
Wilhite, D. A.: Drought as a natural hazard: concepts and definitions, 2000.
Wilhite, D. A. and Glantz, M. H.: Understanding: the Drought Phenomenon: The Role
of Definitions, Water Int, 10, 111–120, https://doi.org/10.1080/02508068508686328,
1107 1985.



Wilhite, D. A. and Pulwarty, R. S.: Drought and Water Crises, edited by: Wilhite, D.
and Pulwarty, R. S., CRC Press, Second edition. | Boca Raton : CRC Press, 2018. | 1st
edition published in 2005., https://doi.org/10.1201/b22009, 2017.
Williams, A. P., Seager, R., Abatzoglou, J. T., Cook, B. I., Smerdon, J. E., and Cook, E.
R.: Contribution of anthropogenic warming to California drought during 2012-2014,
Geophys Res Lett, 42, 6819–6828, https://doi.org/10.1002/2015GL064924, 2015.
Wreford, A. and Neil Adger, W.: Adaptation in agriculture: historic effects of heat
waves and droughts on UK agriculture, Int J Agric Sustain, 8, 278–289,
https://doi.org/10.3763/IJAS.2010.0482, 2010.
Yuan, X., Wang, L., and Wood, E. F.: Anthropogenic Intensification of Southern
African Flash Droughts as Exemplified by the 2015/16 Season, Bull Am Meteorol Soc,
99, S86–S90, https://doi.org/10.1175/BAMS-D-17-0077.1, 2018.
Yuan, X., Wang, L., Wu, P., Ji, P., Sheffield, J., and Zhang, M.: Anthropogenic shift
towards higher risk of flash drought over China, Nature Communications 2019 10:1, 10,
1–8, https://doi.org/10.1038/s41467-019-12692-7, 2019.
Yue, S. and Wang, C. Y.: The Mann-Kendall test modified by effective sample size to
detect trend in serially correlated hydrological series, Water Resources Management,
18, 201–218, https://doi.org/10.1023/B:WARM.0000043140.61082.60, 2004.
Zhang, Q., Kong, D., Singh, V. P., and Shi, P.: Response of vegetation to different time-
scales drought across China: Spatiotemporal patterns, causes and implications, Glob
Planet Change, 152, 1–11, https://doi.org/10.1016/j.gloplacha.2017.02.008, 2017.
Zhao, T. and Dai, A.: The magnitude and causes of global drought changes in the
twenty-first century under a low-moderate emissions scenario, J Clim, 28, 4490–4512,
https://doi.org/10.1175/JCLI-D-14-00363.1, 2015.









