# Peer review of "Distribution, trends and drivers of flash droughts in the United Kingdom."

_EGUsphere, 2024_

## Author Response (AR1)

**Reviewer 1**

**This paper offers an interesting analysis of flash droughts in a region that is often underseen in this context. The text is well-written and presents relevant insights about the emergence and trends of flash droughts in the UK, analysing their relationship with atmospheric and oceanic conditions such as the North Atlantic Oscillation. The authors also present a relevant methodology for identifying the influence of evaporation on the onset of flash droughts. In addition, the authors robustly expose methodological limitations and suggest possible directions for future research based on their findings. However, some justifications and explanations need further clarification.**

Thanks for your positive feedback and the constructive comments, which have contributed to improve the clarity of the manuscript. Herein, we include a detailed response to all your comments.

**General comments:**

**Firstly, the methodology for identifying flash droughts is not sufficiently clear, especially with regard to the application of the SPEI thresholds at the one-month and weekly scales, as well as the maximum duration that a drought event can have to still be considered a flash drought (Lines 193 to 203). I suggest the authors to provide more details on these aspects rather than indicating a citation where these details can be found (Lines 206 to 207).**

Agreed about the missing methodology details. We included additional details to clarify the definition adopted to identify flash drought events as follows:

"*In this study, we identified flash drought events over the UK following the definition suggested by Noguera et al. (2020). For this purpose, we calculated the SPEI at a 1-month time scale and high temporal resolution (i.e., weekly data frequency). Using a short time scale allows for capturing short-term anomalies characteristic of flash drought, while avoiding considering the meteorological anomalies recorded long-term. To identify rapid and anomalous changes in humidity conditions associated with flash droughts onset (Otkin et al., 2018; Svoboda et al., 2002), this method focuses on the identification of quick declines in SPEI values over short periods that reach a certain severity (moderately dry conditions). Thus, a flash drought is defined as a decline in SPEI 1-month values equal to or higher than 2 z-units over a 4-week period that ends in a SPEI value equal to or less than -1.28 z-units (corresponding to a return period of 10 years) (Figure A1). The 4-week period established for the identification of the events, which correspond to the development phase, allows to capture rapid variations in humidity conditions that persist long enough to expect some impact (Noguera et al., 2020), agreeing with some of the most widely used definitions for the assessment of flash droughts (Anderson et al., 2013; Chen et al., 2019; Christian et al., 2019; Osman et al., 2020; Mukherjee and Mishra, 2022). Applying this definition, we identified all flash drought events that occurred in the UK over the period 1969-2021 at seasonal scale (winter: DJF, spring: MAM, summer: JJA, autumn: SON), as well as for the growing-season (MAMJJAS). We assigned flash droughts seasonally based on the week in which their onset was identified.*"

We also included  a diagram in the supplementary material, in response to reviewer #2 request:

[Figure]

**Figure A1.** Diagram of the process followed for the identification of flash droughts.

**Another methodological aspect that I believe is not clear enough and that I have some questions about is the evaluation of the influence of AED on triggering flash droughts. The authors propose comparing the SPEI calculated using conventional time series of precipitation and AED with the SPEI calculated using a fixed AED based on the weekly average. Was it the average weekly AED of the month analysed, the average weekly AED of the year analysed within the period 1969 to 2021, or the weekly average of all the years between 1969 and 2021? (Lines 223 to 236). I wonder if using this methodology, which relies on a SPEI calculated only with a weekly average compared to the conventional SPEI, is sufficiently appropriate to achieve the stated objective. If the AED time series is highly variable, could using averages lead to the emergence of a more influential pattern during periods of more anomalous AED? This could potentially explain, for example, why a clear contribution to the emergence of flash droughts was observed basically only in spring (e.g., Lines 375 to 377, 490 to 491, and 522 to 524). The authors could clarify how this weekly average was used and provide more details about the seasonal variability of the AED time series. Would it be possible to identify the influence of AED on the emergence of flash drought by comparing the SPEI and SPI, both calculated on the same time scale and with the same statistical distribution? Since the SPEI identifies more flash drought events or greater intensities when compared to the SPI, this would be evidence of the AED contribution to the emergence of this type of drought. This approach seems more straightforward and intuitive to me. Could you comment on this?**

Regarding the method used to assess the AED contribution to SPEI;

The weekly average refers to the average of all the years between 1969 and 2021. We clarified this point in the revised manuscript as follows:

 "*while the AED remained at its mean value, which was set at the average AED in each week over the period 1969–2021.*"

The variability of the AED could be a problem for short-term series if these contain several outliers, but we are using long-term series (i.e., 53 years), so it is expected that the average AED value obtained represents well the climate characteristic of the location. Moreover, the hypothesis that higher AED variability could lead to more influential patterns during anomalous periods is not supported by the seasonal AED series (see Figure A8). For example, the variability of AED is much higher in summer than in spring, but in summer there is not a generalised increase in AED contribution. Therefore, it seems reasonable to think that the increase in AED contribution to flash droughts in spring is mainly related to the significant increase in AED observed over the last five decades. Similarly, the only region (i.e., South-East) where a significant increase in AED was recorded in summer is also the only one where a significant increase in AED contribution to flash droughts was observed. Therefore, we believe that the seasonal variability of the AED doesn't affect the reliability of the analyses.

Comparing SPEI and SPI to evaluate AED contribution would be problematic in several ways; On the one hand, the use of the same probability distribution for both indices is not suitable given the typical distribution of precipitation data, which tends to fit better to a Gamma distribution, and climate balance, which tends to fit better to a Log-logistic probability distribution. Precipitation standardisation using two-parameter probability distributions such as Gamma is most appropriate since it has its lower boundary of zero, but climatic balance standardisation requires a three-parameter distribution in which negative values can be present. On the other hand, precipitation series tend to have a higher variability than climatic balance and, consequently, SPI tends to be more variable than SPEI. Thus, although SPEI may indicate a greater severity of drought during warm and dry periods than SPI, the SPI tends to identify a greater number of events [see for example Noguera et al. (2021)] because the methodologies for identifying flash droughts are usually based on rapid variations of a given variable (e.g., precipitation, climatic balance). Therefore, using this comparison as a method to assess AED contribution doesn't seem to be an effective way.

**Would it be possible to compare the flash drought identification results shown in Figures 1 and 2 with historical records of this type of drought? This could work as a way to validate the methodology used. The UKCEH provides a range of information on droughts in the UK on its website (https://www.ceh.ac.uk/our-science/projects/drought-inventory), with data dating back to 1890. For example, within the presented methodology, were more flash droughts identified during years characterized by major drought events in the study area (e.g., 1975-1976, 2004-2006, or 2010-2012)? Could these events be the result of a succession of flash droughts? Or are flash droughts associated with longer drought periods? In this context, it would be interesting if the authors could add a result showing the intra-annual variation in the number of flash drought events.**

This comparison between flash droughts variability and historical drought periods could be interesting, but this would require numerous additional analyses as well as addressing methodological issues in order to make a consistent comparison, so we consider that this is beyond the scope of this study. In fact, this could be a target for future research, which would focus on the transition of these rapidly developing events into long-term events. However, different issues have to be taken into consideration. It should be considered that most of these events of the historical records refer to very long-term droughts (typically 12-month time scales), therefore there will not necessarily be a temporal and spatial coherence between flash droughts and historical droughts periods. In addition, a consistent comparison should include events identified at different time scales (e.g., 3-, 6-, 12-month), and in particular those that may be more representative of agricultural and environmental impacts, which vary

significantly across regions, seasonality and crop/vegetation types. It is true that these dry periods at long-term are more propitious for the occurrence of flash droughts as the recurrence of notable anomalies in precipitation and AED is generally higher, but flash drought frequency depends on how quickly these anomalies occur. Thus, it is expected that the consistency between historical long-term droughts and flash droughts could vary considerably.

To illustrate this issue, we show several examples based on the periods cited by the reviewer. Please find below several examples based on SPI series at 1- and 12-month obtained from the records of the UK Water Resources Portal for different catchments:

[Figure]

**Example 1**: SPI 1- and 12-month between 1975 and 1976 for the Garry catchment (North-West region).

**SPI 1-month**

[Figure]

**SPI 12-month**

**Example 2**: SPI 1- and 12-month between 2004 and 2006 for the Tame catchment (Transition region).

**SPI 1-month**

[Figure]

**SPI 12-month**

[Figure]

**Example 3**: SPI 1- and 12-month between 2010 and 2012 for the Bristol catchment (South-East region).

As it can be observed, these long-term periods characterised by dry conditions usually contain periods of strong negative anomalies developed in the short-term (see months marked with a black dashed line) which could lead to flash droughts. In these particular cases, a certain coherence can be observed, but the frequency of flash drought events during these periods varies notably based on how these anomalies are distributed over the time, including periods with strong short-term anomalies extending over longer periods as well as periods with a succession of strong short-term anomalies combined with wet months. In other cases, 12-month anomalies could be simply the result of slight anomalies accumulated over long periods of time, so the absence of marked anomalies would result in a lack of flash drought events. Thus, the assessment of the connection between historical long-term droughts and flash droughts would require a detailed multiscalar analysis at both regional and seasonal scales.

**The authors highlight that in some parts of the UK a positive trend in the number of flash drought events has been identified. What percentage of the area showed a significant trend? From Figure 4 it appears to be a fairly small and dispersed area compared to the rest of the UK. Could this be related to some bias in the creation of the precipitation and AED products used?**

As the reviewer indicates, the percentage of area showing significant trends in each season is quite small (Winter: 1.2%, Spring: 7.4%, Summer: 1.6%, Autumn: 1.5%, Growing-season: 7.1%). The flash drought increases that we mentioned in the abstract and conclusion are those observed in spring for the whole of the UK (Figure 3). We clarified this point in the revised version of the manuscript.

No spatial bias is observed in the datasets [see additional details in Hollis et al. (2019) and Robinson et al. (2023)]. The absence and dispersion of significant seasonal trends in flash drought variability may be related to the small number of events present in some series. Thus, when a larger set of events is aggregated (e.g., growing-season) the trends are more spatially robust.

**Minor comments:**

**Is AED exactly the same thing as PET? It's not clear (Lines 170 and 171).**

They are not exactly the same but are usually mixed in the literature. Potential evapotranspiration refers to the evaporation under a given meteorological conditions if the water supply was unlimited, but this term may be inappropriate since unlimited soil water would modify the overlaying meteorological conditions [see for example the review provided by Vicente-Serrano et al. (2020)]. In contrast, AED doesn't respond to a land–atmosphere flux but to the demand of water from the atmosphere (i.e., the potential to evaporate water), which is given by a radiative component (i.e., radiation) and aerodynamic component (i.e., air temperature, relative humidity, and wind speed), which is in accordance with Penman-Monteith equation. Indeed, Penman-Monteith equation refer to Reference Evapotranspiration ($ET_0$) since the equation also consider the surface resistance, but in this case given that a similar surface is considered for the entire grid, its variability depends on the capacity of the atmosphere to evaporate and not on the surface resistance, so we believe that the term AED is more appropriate in this case.

**I consider it too strong a statement to say that the influence of AED on triggering flash droughts has been assessed in relation to the global warming process, especially since no analyses of climate change projections or model outputs of this kind have been used (Lines 634 to 635).**

This statement refers to the observed warming over the period analysed, in any case not for the coming years/decades. However, we understand that the sentence can be confusing. We rewrote this sentence in the revised version of the manuscript as follows:

"*We also show the role played by AED on flash drought triggering, as well as its evolution over the last five decades.*"

**Reviewer 2**

This manuscript focuses on flash droughts in the UK. It explains the characteristics and changes of flash droughts from the perspectives of precipitation, atmospheric evaporation demand and circulation conditions. Overall, the structure is clear and the topic is interesting.

There may be issues mainly in the following aspects: 1. The introduction is a bit lengthy and does not seem to clearly outline the research progress of flash droughts (instead of providing a broader overview of drought research in general). 2. The description of the definition of flash droughts is rather brief. I believe the definition of flash droughts might be crucial since there is currently no consistent definition. 3. All the results are qualitative. Although there is extensive discussion, some quantitative results could be incorporated.

Thank you for your positive assessment of the manuscript and the topic addressed, as well as to identify some of the aspects that need to be improved. Herein, we include a detailed response to all your comments.

**Abstract**

**L23: This paper has divided the regions into three zones (North-West, Transition, and South-East). It's best to avoid using other directional terms (e.g., central and northern regions), at least in the Abstract.**

Agreed, we replaced these directional terms in the abstract.

**L27-29: When I read this sentence, I thought this paper was about the relative importance of various factors to flash droughts (e.g., "responds primarily to precipitation variability; AED is important as a secondary driver"). However, the later sections only analyze precipitation and AED.**

We clarified this point in the revised version of the manuscript as follows:

"*We also analysed the relative contribution of the atmospheric evaporative demand (AED) and precipitation to flash drought development in the UK. Our findings show that flash drought occurrence responds primarily to precipitation variability in all seasons, and particularly in winter and autumn.*"

**L31-33: The conclusions can be reconsidered. Does "this relevance" refer to the contribution of AED to flash droughts? I didn't see a correlation analysis. The later sections also don't provide an analysis of how atmospheric and oceanic conditions impact precipitation and evapotranspiration anomalies, only the situation during the onset of flash droughts.**

Yes, we clarified this in the revised version of the manuscript as follows:

"*Moreover, the trends observed in AED contribution evidenced that its relevance is rising significantly in spring and summer in the South-East region, over the study period.*"

Regarding the second point, our aim is not to focus specifically on the impact of atmospheric and oceanic conditions on precipitation and AED as this relationship is well-known in the UK (e.g. the relationship between NAO and precipitation), but rather to illustrate what are the typical conditions preceding events onset in each season. We rewrote this part of the abstract for clarification. In addition, we included a supplementary figure in the review version of the manuscript showing the mean anomalies (%) recorded in precipitation and AED during the weeks preceding top-10 flash drought onset in each season:

[Figure]

**Figure A9.** Seasonal anomalies in precipitation and AED (%) during the development of the top-10 flash droughts of each season over the United Kingdom for the period 1969-2021.

It should be noted that the negative AED anomalies observed in the winter months, although seemingly contradictory, are entirely consistent with the temperature patterns expected during the negative phase of the NAO, which tends to result in dry and cold winters over the UK.

**Introduction**

**The overall structure seems a bit confusing. For example, the second paragraph is lengthy and vaguely introduces current research progress, mentioning the "gap" and then "This study focuses on...," but then expands into several more paragraphs of other research progress.**

**Secondly, most of the progress discussed is about drought rather than flash drought. For instance, the fourth and fifth paragraphs (starting from L100). There are many articles on the impact of precipitation and evapotranspiration on flash droughts and how flash droughts change under climate warming, which may need further review and revision.**

**Thirdly, the statement "no studies previously analysed specifically the occurrence of flash droughts in the UK" is not quite accurate because global or European studies also cover the UK. The review of related progress is lacking.**

We agree with the reviewer's points. We changed the structure of the introduction and included more specific references of flash drought literature. Please find below the new 
[revised manuscript text omitted]

**L196-198: I couldn't understand this definition until I read the reference by Noguera et al. (2020). A more detailed description and a diagram might be needed to clarify this definition.**

We included more details in the revised version of the manuscript to clarify the definition adopted to identify flash drought events:

"*In this study, we identified flash drought events over the UK following the definition suggested by Noguera et al. (2020). For this purpose, we calculated the SPEI at a 1-month time scale and high temporal resolution (i.e., weekly data frequency). Using a short time scale allows for capturing short-term anomalies characteristic of flash drought, while avoiding considering the meteorological anomalies recorded long-term. To identify rapid and anomalous changes in humidity conditions associated with flash droughts onset (Otkin et al., 2018; Svoboda et al., 2002), this method focuses on the identification of quick declines in SPEI values over short periods that reach a certain severity (moderately dry conditions). Thus, a flash drought is defined as a decline in SPEI 1-month values equal to or higher than 2 z-units over a 4-week period that ends in a SPEI value equal to or less than -1.28 z-units (corresponding to a return period of 10 years) (Figure A1). The 4-week period established for the identification of the events, which correspond to the development phase, allows to capture rapid variations in humidity conditions that persist long enough to expect some impact (Noguera et al., 2020), agreeing with some of the most widely used definitions for the assessment of flash droughts (Anderson et al., 2013; Chen et al., 2019; Christian et al., 2019; Osman et al., 2020; Mukherjee and Mishra, 2022). Applying this definition, we identified all flash drought events that occurred in the UK over the period 1969-2021 at seasonal scale (winter: DJF, spring: MAM, summer: JJA, autumn: SON), as well as for the growing-season (MAMJJAS). We assigned flash droughts seasonally based on the week in which their onset was identified.*"

 In addition, we included a diagram in the supplementary material:

[Figure]

**Figure A1.** Diagram of the process followed for the identification of flash droughts.

**L198-L200: Using a four-week window to determine ΔSPEI<-2 only describes the onset of a flash drought. If the onset is very rapid (e.g., within two weeks), it might also "expect some impact." The impact of a flash drought is more related to the entire duration rather than just the onset time (e.g., vegetation might not show negative responses if the flash drought recovers quickly). The description might need to be checked.**

In fact, by setting the 4-week period we are defining a minimum duration, but in this case for the development phase. It is true that there could be cases where a rapid intensification over a 2-week period (e.g., heat waves) would result in impacts, but the number of events in this case without impacts would also be very high if no minimum duration is considered after the onset. The 4-week period established for the development phase is already considered long enough for a response to occur in the different systems potentially affected by drought (e.g., streamflows, soil moisture, crops), regardless of whether wetter conditions occur later.

We illustrate this with this several examples:

[Figure]

**Example 1**: (a) Spatial extension affected by flash drought onset in 1983. (b) Streamflow conditions 4-week before flash drought onset and (c) during flash drought onset.

[Figure]

**Example 2**: (a) Spatial extension affected by flash drought onset in 1990. (b) Streamflow conditions 4-week before flash drought onset and (c) during flash drought onset.

[Figure]

**Example 3**: (a) Spatial extension affected by flash drought onset in 2020. (b) Soil moisture conditions 4-week before flash drought onset and (c) during flash drought onset.

In these examples, which illustrate several flash droughts recorded, we show the variation in flows (1983 and 1990 flash drought) and soil moisture (2020 flash drought, note that soil moisture records from the COSMOS-UK network aren't available for previous events) from the beginning of the decline in SPEI values until flash drought identification (i.e., development phase period). As can be observed, in all cases there is a strong response when the flash drought onset is identified, with notable declines in streamflow and soil moisture. Obviously, it is to be expected that events in which anomalies continue after onset will have greater impacts, but numerous cases such as those shown above could be omitted when these are followed by wet periods (e.g., 1983 flash drought). This would be especially critical in the spring months, when crops are most vulnerable to drought and these dry spells could have an impact even if they do not last over time. It is true that some of these declines occurring over this 4-week period wouldn't have a major impact, as we mentioned in section 4.4 Limitations and future work. Nevertheless, this impact will depend on numerous factors that are difficult to consider (e.g., seasonality, type of vegetation and crops, lithology of the watershed, etc.) as this would require a detailed evaluation of each of the events recorded. Therefore, we believe that the adopted methodology is suitable for flash drought identification, but of course it is difficult to establish a methodology that can be applied on a large scale over different seasons and regions without some limitations.

**L229-L231: I'm not sure if this underestimates the contribution of AED to flash droughts because if precipitation anomalies are sufficient to cause a flash drought, the further exacerbation by AED might be unnoticed.**

Indeed, this is the aim of the methodology, to be able to capture additional severity associated with the increase in AED during flash drought development. The subsequent contribution of the AED to SPEI (i.e., to drought conditions) after flash drought onset was not considered because the objective of this analyst is to assess the contribution of AED only for flash drought development, as this is the phase that distinguishes this type of event from conventional slow-developing events. Thus, we focus on the relative contribution of the AED for the week corresponding to flash drought onset, as SPEI value for this week contains the accumulated anomalies in climatic balance over the last 4 weeks (i.e., flash drought development period considered).

In order to illustrate how the methodology works, we show several examples of flash droughts identified in different locations and seasons:

[Figure]

**Figure.** Relative contribution of AED and precipitation to flash drought development during different flash drought periods.

As it can be observed, SPEI and SPEI_PRE stay quite close as SPEI mainly responds to precipitation, and it is only sensitive to AED during warm and dry periods. Thus, for example, AED contribution to flash drought development in winter in autumn months is close to zero, while in spring and summer the AED can play a crucial role in some events. This is very consistent with the expected seasonal and regional role of the AED triggering drought conditions over the UK, which evidence that the methodology adopted shows coherent contribution percentages.

**L249-L250: It seems flash droughts are identified on individual grid points. How are the top-10 flash droughts selected based on the largest affected area (e.g., number of contiguous grid points)? Also, I'm curious about how seasonal distinctions are made since there could be cross-seasonal flash droughts. How are these divided?**

It is only based on total grid points, there is no continuous area requirement, as drought can occur in different regions at the same time. Regarding the seasonal assignment, we considered

the seasonality of each event based on the week in which the onset is identified. We included these missing details in the revised version of the manuscript:

"*For this purpose, we selected the top-10 flash droughts identified in each season (winter: DJF, spring: MAM, summer: JJA and autumn: SON) for the period 1969-2021 according to the number of the total grid points that recorded flash drought conditions in a given week.*"

"*We assigned flash droughts seasonally based on the week in which their onset was identified.*"

**Results**

**L394-398: I don't quite understand how the changes in precipitation and AED explain the increased contribution of AED to flash drought. The trends of these variables don't seem to match. This might be because flash drought trends are influenced not just by changes in mean precipitation but by variability? I suggest examining how flash droughts change when considering only precipitation (SPEI_PRE in your method)—whether they increase, decrease, or remain unchanged. Then, analyze how they change when considering AED, to explain the relative contribution.**

The reviewer is correct that trends in flash droughts are influenced not only by mean values but also by their variability. However, it is expected that the overall trends of precipitation and AED show certain consistency with the contribution of AED as under increasingly wet conditions (i.e., positive and significant trends in precipitation) the frequency of marked anomalies that result in flash droughts will become less frequent. In this way, and given the metric used for the estimation of the AED contribution (i.e., SPEI), it is necessary to consider that AED contribution not only depends on how high the AED is, but also how low the precipitation is as SPEI sensitivity to AED is mainly limited to dry period. This could seem a limitation, but from an agricultural point of view, an AED increase wouldn't have a relevant role triggering drought conditions if precipitation is above or around mean values. In fact, under these conditions, an increase in AED would favour vegetation growth.

Regarding the suggestion of evaluating the AED contribution based on the differences in the number of events, this approach would present the problem mentioned by the reviewer in the comment of the **L229-L231**. On the one hand, comparing the number of events would not allow estimating the additional AED because as long as the precipitation deficit is sufficient to reach the established thresholds, it does not matter how high the AED is, as the number of events would be the same. On the other hand, this method would not allow estimating the contribution as an independent variable, so it would be necessary to consider the contribution based on the difference in the number of events, which is a rather crude approximation. In addition, considering the low percentages of the contribution of the AED to flash drought observed and the high correlation between SPEI and SPEI PRE, it is expected that the number of events identified by SPEI and SPEI PRE would be quite similar, with some exceptions in summer over South-East regions. To illustrate this issue, we show several examples of flash drought identification by means of both versions of the SPEI:

[Figure]

**Example 1**. SPEI (red line) and SPEI PRE (blue line) series 1975-1976 for a random point in North-West region. Red and blue points represent the flash drought onset identified by SPEI and SPEI PRE, respectively.

[Figure]

**Example 2**. SPEI (red line) and SPEI PRE (blue line) series 2001-2003 for a random point in Transition region. Red and blue points represent the flash drought onset identified by SPEI and SPEI PRE, respectively.

[Figure]

**Example 3**. SPEI (red line) and SPEI PRE (blue line) series 2004-2007 for a random point in South-East region. Red and blue points represent the flash drought onset identified by SPEI and SPEI PRE, respectively.

These examples show SPEI and SPEI_PRE series for a random point of each region (North-West, Transition, South-East) for different periods over 1969-2021. Red (SPEI) and blue (SPEI_PRE) points show the weeks in which flash drought are identified, and as it can be observed, the events identified are quite similar. Although just short periods are shown in the graph for better visualitation, the total number of events identified by both metrics over the whole period in the tree examples is quite similar (±3 events), and the correlation between SPEI and SPEI PRE series is very high (0.99 in North-West example, 0.98 in Transition example, and 0.97 in South-East example according to Pearson's test). We included a couple of flash drought events identified by SPEI but not for SPEI PRE (see Example 3). Thus, the main differences between SPEI and SPEI PRE series are observed in the South-East region, as it is the region in which AED is more relevant in triggering drought conditions, but there is also a high consistency between SPEI and SPEI PRE even in this region.

**Discussion**

**L519-520: Please add figure citations (e.g., Figure 1) to the relevant conclusions.**

Done.

**####Line Edits####**

**L20: affected -> have affected (consistent tense)**

**L31: evidenced -> evidence**

**L91: delete " ; "**

**L165: The full name of PET is potential evapotranspiration**

**L369: No need to define the abbreviation again**

**L408: Change to "is only"**

**L435: develop -> development, ocurred -> occurred**

Thanks for the corrections. These were included in the revised manuscript.

---

## Author Response (AR2)

Iván Noguera

UK Centre for Ecology & Hydrology

ivanog@ceh.ac.uk

Dear Editor of Hydrology and Earth System Sciences,

Thank you for accepting the article as is. Therefore, no relevant changes have been made in the manuscript, only some grammatical enhancement, correction of typos and improvement of the appearance of some figures.

Best regards,

Ivan

Address for correspondence: UK Centre for Ecology & Hydrology, Benson Lane, Maclean Building, Crowmarsh Gifford, Wallingford OX10 8BB, Oxfordshire, UK.